# Node-wise Filtering in Graph Neural Networks: A Mixture of Experts Approach

## Abstract

Graph Neural Networks (GNNs) have shown strong performance in node classification across diverse graph structures. Most GNNs apply a uniform global filter—typically low-pass for homophilic graphs and high-pass for heterophilic ones. However, real-world graphs often exhibit a complex mix of both patterns, making a single global filter suboptimal. While a few recent methods incorporate multiple filters, they typically apply all filters across all nodes, leading to both inefficiency and limited adaptivity. Moreover, these approaches often lack theoretical grounding for when and why such multi-filter strategies work. In this work, we provide theoretical analysis showing that global filters optimized for one structural pattern can harm performance on others, and we characterize conditions under which node-wise filtering achieves optimal separability. Building on these insights, we propose NODE-MOE, a novel mixture-of-experts GNN framework that adaptively selects filters for each node. Extensive experiments across homophilic and heterophilic graphs demonstrate the effectiveness and efficiency of NODE-MOE.

## 1 Introduction

Graph Neural Networks (GNNs) (Kipf & Welling, 2016) have emerged as powerful tools in representation learning for graph structure data, and have achieved remarkable success on various graph learning tasks (Wu et al., 2020; Ma & Tang, 2021), especially the node classification task. GNNs usually can be designed and viewed from two domains, i.e., spatial domain and spectral domain. In the spatial domain, GNNs (Kipf & Welling, 2016; Hamilton et al., 2017; Gasteiger et al., 2018) typically follow the message passing mechanism (Gilmer et al., 2017), which propagate messages between neighboring nodes. In the spectral domain, GNNs (Defferrard et al., 2016; Chien et al., 2020) apply different filters on the graph signals.

Most GNNs have shown great effectiveness in the node classification task of homophilic graphs (Wu et al., 2019; Gasteiger et al., 2018; Baranwal et al., 2021), where connected nodes tend to share the same labels. These GNNs usually leverage the low-pass filters, where the smoothed signals are preserved. However, the heterophilic graphs exhibit the heterophilic patterns, where the connected nodes tend to have different labels. As a result, several GNNs (Sun et al., 2022; Li et al., 2024; Bo et al., 2021) designed for heterophilic graphs introduce the high-pass filter to better handle such diversity. To adapt to both homophilic and heterophilic graphs, GNNs with learnable graph convolution (Chien et al., 2020; Bianchi et al., 2021; He et al., 2021; 2022) can automatically learn different types of filters for different types of graphs. Despite the great success, these GNNs usually apply a uniform global filter across all nodes.

However, real-world graphs often display a complex interplay of homophilic and heterophilic patterns (Luan et al., 2022; Mao et al., 2024), challenging this one-size-fits-all filtering approach. Specifically, while some nodes tend to connect with others that share similar labels, reflecting homophilic patterns, others are more inclined to form connections with nodes that have differing labels, indicative of heterophilic patterns. Several methods, such as ACM-GNN (Luan et al., 2022), AutoGCN (Wu et al., 2022), PC-Conv (Li et al., 2024) and ASGAT (Li et al., 2021) leverage different global filters to alleviate this issue. These methods, referred to as post-fusion methods, apply multiple filters to all nodes and subsequently combine the predictions of different filters using attention or learned weighting mechanisms. While this allows for some adaptivity, it remains inefficient, as all filters are evaluated for all nodes regardless of relevance. Moreover, the combination process typically relies solely on node representations, ignoring local structural cues such as neighborhood heterogeneity, and the resulting weights often lack theoretical justification—raising concerns about

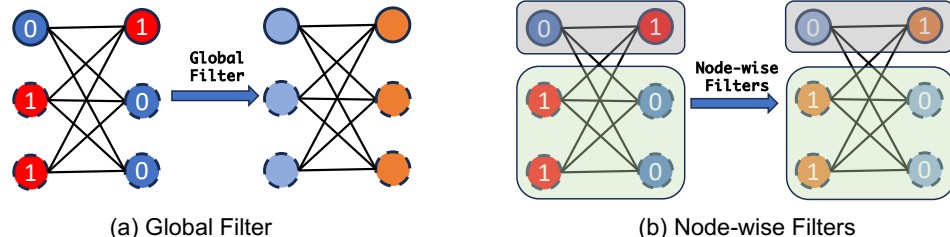

(a) Global Filter             (b) Node-wise Filters

Figure 1: A toy example illustrating the effect of global vs. node-wise filtering. Node colors represent features, and numbers indicate class labels. Solid-edge nodes follow homophilic patterns, with most neighbors sharing the same label (2 out of 3), while dashed-edge nodes follow heterophilic patterns, with most neighbors having different labels.

their reliability in distinguishing between homophilic and heterophilic nodes. Besides, reusing the same filter set for all nodes can lead to potential issues.

To illustrate the limitations of global filtering, consider the toy example in Figure 1(a), where node colors denote features and numbers indicate labels. Solid and dotted circles represent nodes following homophilic and heterophilic patterns, respectively. Applying a global low-pass filter $1 - \lambda$, where $\lambda$ is the eigenvalue of graph Laplacian matrix, uniformly across all nodes causes nodes on the left to converge to one feature value and those on the right to another, despite having different labels, resulting in misclassification. The indistinguishability of the filtered features can further impact post-fusion methods, as these methods rely on those features to compute filter weights.

This toy example clearly illustrates the limitations of one-size-fits-all filtering and highlights the need for a more adaptive approach. Rather than applying a fixed set of filters to all nodes, we advocate for a node-wise filtering strategy that assigns different filters to different nodes based on their local structural patterns. As shown in Figure 1(b), applying a low-pass filter (e.g., $1 - \lambda$) to homophilic nodes and a high-pass filter (e.g., $\lambda - 1$) to heterophilic nodes results in nodes from the same class sharing similar features. This leads to perfect classification in the example and demonstrates the potential of structure-aware, node-wise filtering.

In this work, we observe that while many real-world graphs contain diverse structural patterns, these patterns often vary significantly across different communities within the same graph. Using the Contextual Stochastic Block Model (CSBM) to simulate graphs with mixed patterns, we theoretically show that global filters optimized for a specific structural pattern can lead to significant performance degradation on mismatched patterns. In contrast, node-wise filtering guided by structural cues enables linear separability under mild conditions and offers a principled way to adapt filters based on local graph structure. Motivated by these findings, we propose NODE-MOE, a node-wise filtering framework based on a Mixture-of-Experts architecture that adaptively selects appropriate filters for each node. Extensive experiments across both homophilic and heterophilic graphs demonstrate the effectiveness and interpretability of NODE-MOE.

## 2 PRELIMINARY

In this section, we first introduce the notations and background concepts used throughout the paper. We then examine the structural patterns in real-world graph datasets, which often exhibit a mix of homophilic and heterophilic connections. Finally, we present a theoretical analysis showing that global filters tend to fail in such mixed-structure settings, while node-wise filtering can achieve linear separability under mild assumptions.

**Notations.** We use bold upper-case letters such as $\mathbf{X}$ to denote matrices. $\mathbf{X}_i$ denotes its $i$-th row and $\mathbf{X}_{ij}$ indicates the $i$-th row and $j$-th column element. We use bold lower-case letters such as $\mathbf{x}$ to denote vectors. Let $\mathcal{G} = (\mathcal{V}, \mathcal{E})$ be a graph, where $\mathcal{V}$ is the node set, $\mathcal{E}$ is the edge set, and $|\mathcal{V}| = n$. $\mathcal{N}_i$ denotes the neighborhood node set for node $v_i$. The graph can be represented by an adjacency matrix $\mathbf{A} \in \mathbb{R}^{n \times n}$, where $\mathbf{A}_{ij} > 0$ indices that there exists an edge between nodes $v_i$ and $v_j$ in $\mathcal{G}$, or otherwise $\mathbf{A}_{ij} = 0$. For a node $v_i$, we use $\mathcal{N}(v_i) = \{v_j : \mathbf{A}_{ij} > 0\}$ to denote its neighbors. Let $\mathbf{D} = diag(d_1, d_2, \ldots, d_n)$ be the degree matrix, where $d_i = \sum_j \mathbf{A}_{ij}$ is the degree of node $v_i$. Furthermore, suppose that each node is associated with a $d$-dimensional feature $\mathbf{x}$ and we use $\mathbf{X} = [\mathbf{x}_1, \ldots, \mathbf{x}_n]^\top \in \mathbb{R}^{n \times d}$ to denote the feature matrix. Besides, the label matrix is $\mathbf{Y} \in \mathbb{R}^{n \times c}$, where $c$ is the number of classes. We use $y_v$ to denote the label of node $v$.

**Graph Laplacian.** The graph Laplacian matrix is defined as $\mathbf{L} = \mathbf{D} - \mathbf{A}$. We define the normalized adjacency matrix as $\tilde{\mathbf{A}} = \mathbf{D}^{-\frac{1}{2}}\mathbf{A}\mathbf{D}^{-\frac{1}{2}}$ and the normalized Laplacian matrix as $\tilde{\mathbf{L}} = \mathbf{I} - \tilde{\mathbf{A}}$. Its eigendecomposition can be represented by $\tilde{\mathbf{L}} = \mathbf{U}\mathbf{\Lambda}\mathbf{U}^{\top}$, where the $\mathbf{U} \in \mathbb{R}^{n \times n}$ is the eigenvector matrix and $\mathbf{\Lambda} = diag(\lambda_1, \lambda_2, \ldots, \lambda_n)$ is the eigenvalue matrix. Specifically, $0 \le \lambda_1 \le \lambda_2 \le \cdots \le \lambda_n < 2$. The filtered signals can be represented by $\hat{\mathbf{X}} = \mathbf{U}f(\mathbf{\Lambda})\mathbf{U}^{\top}\mathbf{X}$, where $f$ is the filter function. As a result, the graph convolution $\tilde{\mathbf{A}}\mathbf{X}$ can be viewed as a low-pass filter, with the filter $f(\lambda_i) = 1 - \lambda_i$. Similarly, the graph convolution $-\tilde{\mathbf{A}}\mathbf{X}$ is a high-pass filter with filter $f(\lambda_i) = \lambda_i - 1$.

**Homophily metrics.** Homophily metrics measure the tendency of edges to connect nodes with similar labels (Platonov et al., 2024). There are several commonly used homophily metrics, such as edge homophily (Zhu et al., 2020), node homophily (Pei et al., 2020), and class homophily (Lim et al., 2021b). In this paper, we adopt the node homophily $H(\mathcal{G}) = \frac{1}{|\mathcal{V}|}\sum_{v_i \in \mathcal{V}} h(v_i)$, where $h(v_i) = \frac{|\{u \in \mathcal{N}(v_i): y_u = y_v\}|}{d_i}$ measures the label similarity between node $v_i$ with its neighbors. A node with higher $h(v)$ exhibits a homophilic pattern while a low $h(v)$ indicates a heterophilic pattern.

## 2.1 STRUCTURAL PATTERNS IN EXISTING GRAPHS

Real-world graphs often exhibit a mixture of homophilic and heterophilic patterns (Luan et al., 2022; Mao et al., 2024). While most nodes in a graph may follow one predominant pattern, some nodes exhibit the opposite. Building on these observations, we further examine how such structural patterns vary across different regions of the same graph. Specifically, we analyze two homophilic datasets, Cora and CiteSeer (Sen et al., 2008), and two heterophilic datasets, Chameleon and Squirrel (Rozemberczki et al., 2021). We apply a community detection algorithm (Fortunato, 2010) to partition each graph and compute the homophily level within the largest 10 communities.

As shown in Figure 2, we sort communities in each graph by decreasing homophily and observe **substantial variation in homophily levels across communities within the same graph**. For example, in the Cora dataset, some communities exhibit strong homophily (close to 1), while others fall below 0.5. Similarly, in the Chameleon dataset, homophily ranges from near 0 to above 0.6. These findings not only underscore the presence of mixed structural patterns, but also reveal that these patterns are distributed non-uniformly across different regions of the graph.

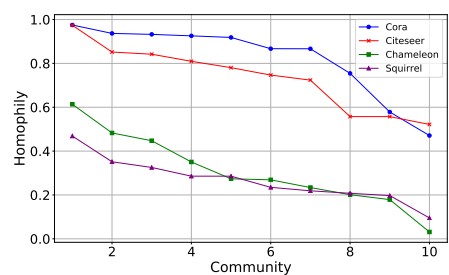

Figure 2: Homophily in different communities.

## 2.2 ANALYSIS BASED ON CSBM MODEL

To further analyze the impact of mixed structural patterns, we employ the Contextual Stochastic Block Model (CSBM)(Deshpande et al., 2018), a widely used generative framework in graph analysis(Fortunato & Hric, 2016; Jiang et al., 2023) and GNN studies (Palowitch et al., 2022; Baranwal et al., 2021; Ma et al., 2021), to construct graphs that explicitly contain both homophilic and heterophilic patterns. CSBM typically assumes a uniform structural pattern—nodes with the same label connect with probability $p$, and nodes with different labels connect with probability $q$ (Ma et al., 2021). To reflect the coexistence of diverse patterns in real-world graphs, we adapt the CSBM by combining two distinct instances, one homophilic and one heterophilic, into a single graph, similar to the approach of Mao et al. (2024).

**Definition 1.** $CSBM(n, \boldsymbol{\mu}, \boldsymbol{\nu}, (p_0, q_0), (p_1, q_1), P)$. *For a graph, the generated nodes consist of two classes, $C_0 = \{i \in [n] : y_i = 0\}$ and $C_1 = \{j \in [n] : y_j = 1\}$. For each node, consider $\mathbf{X} \in \mathbb{R}^{n \times d}$ to be the feature matrix such that each row $\mathbf{X}_i$ is an independent $d$-dimensional Gaussian random vectors with $\mathbf{X}_i \sim N\left(\boldsymbol{\mu}, \frac{1}{d}\mathbf{I}\right)$ if $i \in C_0$ and $\mathbf{X}_j \sim N\left(\boldsymbol{\nu}, \frac{1}{d}\mathbf{I}\right)$ if $j \in C_1$. Here $\boldsymbol{\mu}, \boldsymbol{\nu}$ are the fixed class mean vectors with $\|\boldsymbol{\mu}\|_2, \|\boldsymbol{\nu}\|_2 \le 1$ and $\mathbf{I}$ is the identity matrix. Suppose there are two patterns of nodes in the adjacency matrix $\mathbf{A} = (a_{ij})$, i.e., the homophilic pattern: $H_0 = \{i \in [n] : a_{ij} = \text{Ber}(p_0) \text{ if } y_i = y_j \text{ and } a_{ij} = \text{Ber}(q_0) \text{ if } y_i \ne y_j, p_0 > q_0\}$ and the heterophilic pattern: $H_1 = \{i \in [n] : a_{ij} = \text{Ber}(p_1) \text{ if } y_i = y_j \text{ and } a_{ij} = \text{Ber}(q_1) \text{ if } y_i \ne y_j, p_1 < q_1\}$. $P$ denotes the probability that a node is in the homophilic pattern. We also assume the nodes follow the same degree distribution $p_0 + q_0 = p_1 + q_1$.*

For simplification, we consider a linear model with parameters $\mathbf{w} \in \mathbb{R}^d$ and $b \in \mathbb{R}$, following the previous method (Baranwal et al., 2021). The predicted label for nodes is given by $\hat{\mathbf{y}} = \sigma(\tilde{\mathbf{X}}\mathbf{w} + b\mathbf{1})$, where $\sigma$ is the sigmoid function, and $\tilde{\mathbf{X}}$ represents the features after filtering. The binary cross-entropy loss over nodes $\mathcal{V}$ is formulated as $L(\mathcal{V}, \mathbf{w}, b) = -\frac{1}{|\mathcal{V}|} \sum_{i \in \mathcal{V}} y_i \log(\hat{y}_i) + (1 - y_i) \log(1 - \hat{y}_i)$.

**Theorem 1.** *Suppose $n$ is relatively large, the graph is not too sparse with $p_i, q_i = \omega(\log^2(n)/n)$ and the feature center distance is not too small with $\|\boldsymbol{\mu} - \boldsymbol{\nu}\| = \omega(\frac{\log n}{\sqrt{\mathrm{dn}(p_0+q_0)}})$ and $\|\mathbf{w}\| \leq R$.*

*For the graph $G(\mathcal{V}, \mathcal{E}, \mathbf{X}) \sim CSBM(n, \boldsymbol{\mu}, \boldsymbol{\nu}, (p_0, q_0), (p_1, q_1), P)$: If the low-pass global filter, i.e., $1 - \lambda$, is applied to the whole graph $G$, we can find a optimal $\mathbf{w}^*, b^*$ that achieve near linear separability for the homophilic node set $H_0$. However, the loss for the heterophilic node set $H_1$ can be relatively large with:*

$$L(H_1, \mathbf{w}^*, b^*) \geq \frac{R(q_1 - p_1)}{2(q_1 + p_1)} \|\boldsymbol{\mu} - \boldsymbol{\nu}\| \left(1 + o_d(1)\right).$$

The proof is provided in Appendix A.1. Theorem 1 shows that applying a global low-pass filter can yield a near-optimal classifier for homophilic nodes, achieving linear separability. However, it also results in substantial loss for heterophilic nodes, underscoring the limitations of a uniform global filtering strategy in graphs with mixed structural patterns.

**Theorem 2.** *Under the same assumptions as Theorem 1, if different filters are applied to homophilic and heterophilic sets separately, we can find an optimal $\mathbf{w}^*, b^*$ that all the nodes are linear separable with the probability:*

$$\mathbb{P}\left(\left(\tilde{\mathbf{X}}_i\right)_{i \in \mathcal{V}} \text{ is linearly separable }\right) = 1 - o_d(1).$$

The proof can be found in Appendix A.2. Theorem 2 demonstrates that applying different filters to homophilic and heterophilic nodes separately enables linear separability across all nodes, highlighting the advantage of node-wise filtering strategies.

**Theorem 3.** *Under the same assumptions as Theorem 1, homophilic and heterophilic nodes can be separated based on the feature distance between a node and the average feature vector of its neighbors, given by $\|\mathbf{X}_i - \sum_{j \in \mathcal{N}(i)} \frac{\mathbf{X}_j}{D_{ii}}\|$ with probability $P = 1 - o_d(1)$.*

The proof is provided in Appendix A.3. Theorem 3 reveals that homophilic and heterophilic nodes can be effectively distinguished based on their local structure, using the feature distance between a node and the average of its neighbors' features. These findings strongly motivate the development of a node-wise filtering approach that can automatically assign different filters to individual nodes based on their local structural patterns, thereby improving overall model performance.

## 3 THE PROPOSED METHOD

In this section, we aim to design a model that addresses the limitations of global filtering in graphs with mixed structural patterns based on our empirical observations and theoretical analysis.

### 3.1 NODE-MOE: NODE-WISE FILTERING VIA MIXTURE OF EXPERTS

Mixture of Experts (MoE) (Jacobs et al., 1991; Jordan & Jacobs, 1994), which follows the divide-and-conquer principle to divide the complex problem space into several subspaces so that each one can be easily addressed by specialized experts, have been successfully adopted across various domains (Masoudnia & Ebrahimpour, 2014; Shazeer et al., 2017; Riquelme et al., 2021). For node classification tasks in graphs exhibiting a mixture of structural patterns, the diversity of node interactions necessitates applying distinct filters to different nodes as we discussed in Sections 2. This necessity aligns well with the MoE methodology, which processes different samples with specific experts. Building on this principle, we introduce a flexible and efficient Node-wise Filtering via Mixture of Experts (NODE-MOE) framework, designed to dynamically apply appropriate filters to nodes based on their structural characteristics.

The overall NODE-MOE framework is illustrated in Figure 3, which consists of two primary components: the gating model and the multiple expert models. With the graph data as input, the gating model $g(\cdot)$ computes the weight assigned to each expert for every node, reflecting the relevance of each expert's contribution to that specific node. Each expert model, implemented as any GNN

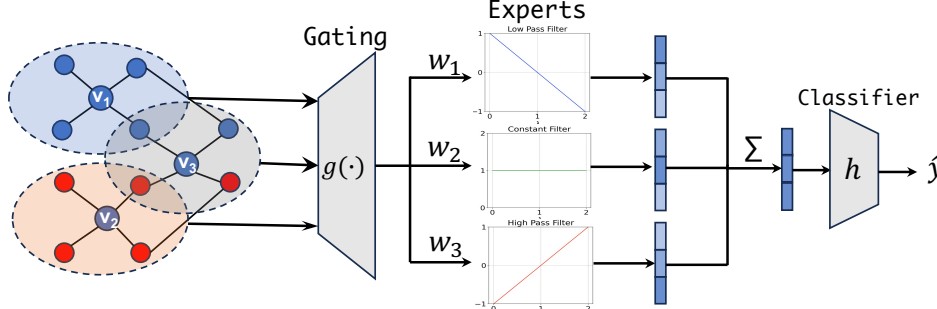

Figure 3: Overall framework of the proposed NODE-MOE. For each node, the gating model selects different experts based on the node's features and local structure.

with different filters, generates node representations independently. The final node classification is determined by a weighted sum of these representations, where the weights are those assigned by the gating model. The prediction for node $i$ can be represented by:

$$\hat{y}_i = \text{Classifier} \left( \sum_{o=1}^{m} g(\mathbf{A}, \mathbf{X})_{i,o} E_o(\mathbf{A}, \mathbf{X})_i \right), \qquad (1)$$

where $m$ is the number of experts, $E_o$ denotes the $o$-th expert, and $g(\mathbf{A}, \mathbf{X})_{i,o}$ represents the weight assigned to the $o$-th expert for node $i$ by the gating model. In the following, we will delve into the specific designs of the gating model and the expert models.

## 3.2 GATING MODEL

The gating model is a pivotal component of the Node-MoE framework, aimed at selecting the most appropriate experts for each node. Its primary function is to dynamically assign higher weights to experts whose filtering characteristics best match the node's patterns. For instance, an expert utilizing a high-pass filter may receive a higher weight for a node that exhibits heterophilic patterns. However, a significant challenge arises as there is no explicit ground truth indicating which pattern each node belongs to. In traditional MoE models, the gating model often utilizes a straightforward feed-forward network that processes the features of the sample as its input (Shazeer et al., 2017; Riquelme et al., 2021; Du et al., 2022; Wang et al., 2024). Nevertheless, the nodes with different patterns may share similar node features after filtering as shown in Figure 1, making this method ineffective.

To address this challenge, we estimate node patterns by incorporating the contextual features surrounding each node inspired by Theorem 3. If a node's features significantly differ from those of its neighboring nodes, it is likely that this node exhibits a heterophilic pattern. Specifically, the input to our gating model includes a composite vector $[\mathbf{X}, |\mathbf{A}\mathbf{X} - \mathbf{X}|, |\mathbf{A}^2\mathbf{X} - \mathbf{X}|]$. This vector combines the node's original features with the absolute differences between its features and those of its neighbors over one and two hops, respectively, to indicate the node's structural patterns. Moreover, as discussed in Section 2.1, different structural patterns are not uniformly distributed across the graph, and distinct communities may exhibit varying structural characteristics. To capitalize on this phenomenon, we employ GNNs with low-pass filters, such as GIN (Xu et al., 2018), for the gating model. These networks are chosen due to their strong community detection capabilities (Shchur & Günnemann, 2019; Bruna & Li, 2017), ensuring that neighboring nodes are likely to receive similar expert selections. Experimental results in Section 4.3 clearly demonstrate the proposed gating can efficiently assign different nodes to their suitable filters. We also compare the effectiveness of different gating backbone models in Appendix D.2

## 3.3 EXPERT MODELS

The mixed structural patterns found in real-world graphs require that the expert models in NODE-MOE exhibit diverse capabilities. To this end, we incorporate multiple GNNs equipped with different filters. While traditional GNNs often rely on fixed filters, these may fall short in capturing the complexity of diverse structural patterns. Instead, we employ GNNs with learnable convolutions (Chien et al., 2020; Bianchi et al., 2021; He et al., 2021; 2022), which adapt their filters to better fit varying graph patterns. However, using identical experts can hinder the gating model's ability to differentiate between them (Chen et al., 2022) and may lead all filters to converge similarly. To promote diversity and ensure each expert specializes in distinct structural patterns, we adopt a differentiated initialization

strategy. Specifically, we initialize the experts with distinct types of filters, such as low-pass, constant, and high-pass. More implementation details are provided in Section 4.

**Filter Smoothing Loss.** While integrating multiple experts with diverse filters significantly enhances the expressive capacity of our NODE-MOE framework, this complexity can also make the model more challenging to fit. For example, training multiple filters simultaneously may lead to oscillations in the spectral domain for each filter as shown in Appendix B. This not only makes the model overfitting but also impacts its explainability. To mitigate these issues, we introduce a filter smoothing loss designed to ensure that the learned filters exhibit smooth behavior in the spectral domain. Specifically,

$$L_s^o = \sum_{i=1}^{K} |f_o(s_i) - f_o(s_{i-1})|^2, \tag{2}$$

where $f_o(\cdot)$ is the learnable filter function of the $o$-th expert, $s_0 \leq s_1 \leq \cdots \leq s_K$ are $K+1$ values spanning the spectral domain. By minimizing the activation differences between neighboring values in the spectral domain, the filter functions become smoother. The overall training loss is then given by $L = L_{task} + \gamma \sum_{o=1}^{m} L_s^o$, where the $L_{task}$ is the node classification loss and $\gamma$ is a hyperparameter that adjusts the influence of the filter smoothing loss.

### 3.4 TOP-K GATING

The soft gating that integrates all experts in the Node-MoE framework significantly enhances its modeling capabilities, but it also increases computational complexity since each expert must process all samples. To improve computational efficiency while maintaining performance, we introduce a variant of NODE-MOE by leveraging the Top-K gating mechanism. In this variant, NODE-MOE with Top-K gating selectively activates only the top k experts with the highest relevance for each node. Specifically, the gating function for a node $v_i$ is defined as $g(v_i) = \text{Softmax}\left(\text{TopK}\left(g\left(\mathbf{A}, \mathbf{X}\right)_i, k\right)\right)$. To prevent the gating model from consistently favoring a limited number of experts, we also incorporate the load-balancing loss (Shazeer et al., 2017), which encourage the gating model assign similar number of samples to each expert.

### 3.5 TIME COMPLEXITY OF NODE-MOE

The time complexity of the proposed NODE-MOE can be significantly reduced through sparse Top-K gating. For instance, when setting $K = 1$, each node only needs to be processed by a single expert. In this case, the time complexity of NODE-MOE becomes comparable to that of a single expert, with the addition of a lightweight gating model. In the section 4.4, we demonstrate the effective and efficiency of the proposed NODE-MOE.

## 4 EXPERIMENT

In this section, we conduct comprehensive experiments to evaluate the effectiveness and efficiency of the proposed NODE-MOE. We assess its performance, analyze the behavior of the expert selection mechanism, and examine the influence of key design choices through ablation studies.

### 4.1 EXPERIMENTAL SETTINGS.

**Datasets.** To evaluate the efficacy of NODE-MOE, we conduct experiments across seven widely used datasets. These include four homophilic datasets: Cora, CiteSeer, Pubmed (Sen et al., 2008), and ogbn-arxiv (Hu et al., 2020); along with four heterophilic datasets: Chameleon, Squirrel (Pei et al., 2020), Penn94 and pokec Lim et al. (2021a). For Cora, CiteSeer, and Pubmed, we generate 10 random splits, distributing nodes into 60% training, 20% validation, and 20% testing partitions. For the heterophilic datasets, we utilize the 10 fixed splits as specified in Pei et al. (2020) and Lim et al. (2021a). The ogbn-arxiv dataset is evaluated using its standard split (Hu et al., 2020). We run the experiments 3 times for each split and report the average performance and standard deviation. More details about these datasets are shown in Appendix C.1. We also evaluate on more heterophilic datasets, i.e., amazon-ratings and tolokers (Platonov et al., 2023), with results in Appendix C.2.

**Baselines.** We compare our method with a diverse set of baselines, which can be divided into five categories: (1) Non-GNN methods like MLP and Label Propagation (LP) (Zhou et al., 2003); (2) Homophilic GNNs utilizing fixed low-pass filters such as GCN (Kipf & Welling, 2016), GAT (Veličković et al., 2017), APPNP (Gasteiger et al., 2018), and GCNII (Chen et al., 2020); (3) Heterophilic GNNs including AutoGCN (Wu et al., 2022), WRGCN (Suresh et al., 2021), PC-Conv (Li et al., 2024), ACM-GCN (Luan et al., 2022), ASGAT (Li et al., 2021) and LinkX (Lim et al., 2021a); (4) GNNs with learnable filters like GPRGNN (Chien et al., 2020) and ChebNetII (He et al., 2022); (5) MoE-based GNNs such as GMoE (Wang et al., 2024) and Mowst (Zeng et al., 2023).

Table 1: Node classification accuracy (%) on benchmark datasets. OOM means out-of-memory. The bold and underline markers denote the best and second-best performance, respectively. *indicates a t-test with $p < 0.05$.

| Methods | Homophilic datasets | | | | Heterophilc datasets | | | |
|---|---|---|---|---|---|---|---|---|
| | Cora | CiteSeer | PubMed | ogbn-arxiv | Chameleon | Squirrel | Penn94 | Pokec |
| MLP | 76.49 ± 1.13 | 73.15 ± 1.36 | 86.14 ± 0.64 | 55.68 ± 0.11 | 48.11 ± 2.23 | 31.68 ± 1.90 | 73.61±0.40 | 62.39 ± 0.06 |
| LP | 86.05 ± 1.35 | 69.39 ± 2.01 | 83.38 ± 0.64 | 68.14 ± 0.00 | 44.10 ± 4.10 | 31.92 ± 0.82 | 63.26 ± 0.41 | 53.28 ± 0.05 |
| GCN | 88.60 ± 1.19 | 76.88 ± 1.78 | 88.48 ± 0.46 | 71.91 ± 0.15 | 67.96 ± 1.82 | 54.47 ± 1.17 | 82.37 ± 0.24 | 75.43 ± 0.15 |
| GAT | 88.68 ± 1.13 | 76.70 ± 1.81 | 86.52 ± 0.56 | 71.92 ± 0.17 | 65.29 ± 2.54 | 49.46 ± 1.69 | 81.53 ± 0.55 | 71.77 ± 6.18 |
| APPNP | 88.49 ± 1.28 | 77.42 ± 1.47 | 87.56 ± 0.52 | 71.61 ± 0.30 | 54.32 ± 2.61 | 36.41 ± 1.94 | 74.33 ± 0.38 | 62.58 ± 0.08 |
| GCNII | 88.12 ± 1.05 | 77.30 ±1.58 | 90.17 ± 0.57 | 72.74 ± 0.16 | 55.54 ± 2.02 | 56.63 ± 1.17 | 82.92±0.59 | 78.94 ± 0.11 |
| AutoGCN | 87.59 ± 1.17 | 75.12 ± 1.94 | 89.13 ± 0.51 | 69.34 ± 0.63 | 65.21 ± 2.97 | 45.55 ± 1.54 | 81.02 ± 0.16 | 79.49 ± 0.33 |
| WRGCN | 88.06 ± 1.50 | 76.28 ± 1.98 | 86.39 ± 0.55 | >24h | 65.24 ± 0.87 | 48.85 ± 0.78 | 75.50 ± 0.09 | >24h |
| PC-Conv | 88.85 ± 1.29 | 77.30 ± 1.79 | 85.79 ± 0.64 | 67.21 ± 0.19 | 66.86 ± 1.97 | 44.75 ± 1.58 | 85.36 ± 0.06 | 77.86 ± 0.07 |
| ACMGCN | 88.01 ± 1.26 | 76.52 ± 1.72 | 89.51 ± 0.49 | 62.09 ± 1.29 | 69.62 ± 1.22 | 57.02 ± 0.79 | 83.02 ± 0.65 | 74.13 ± 0.14 |
| ASGAT | 86.63 ± 1.51 | 73.76 ± 1.17 | OOM | OOM | 66.50 ± 2.80 | 55.80 ± 3.20 | OOM | OOM |
| LinkX | 82.89 ± 1.27 | 70.05 ± 1.88 | 84.81 ± 0.65 | 66.54 ± 0.52 | 68.42 ± 1.38 | 61.81 ± 1.80 | 84.71 ± 0.52 | 81.86 ± 0.21 |
| GPR-GNN | 88.54 ± 0.67 | 76.44 ± 1.89 | 88.46 ± 0.31 | 71.78 ± 0.18 | 62.85 ± 2.90 | 54.35 ± 0.87 | 83.54 ± 0.32 | 80.74±0.22 |
| ChebNetII | 88.71 ± 0.93 | 76.93 ± 1.57 | 88.93 ± 0.29 | 72.32 ± 0.23 | 71.14 ± 2.13 | 57.12 ± 1.13 | 84.86 ± 0.33 | 81.16 ± 0.04 |
| GMoE | 87.27 ± 1.74 | 76.56 ± 1.57 | 88.14 ± 0.56 | 71.74 ± 0.29 | 71.88 ± 1.60 | 51.97 ± 3.16 | 75.76 ± 4.39 | 59.30 ± 1.92 |
| Mowst | 86.18 ± 1.45 | 75.27 ± 2.19 | 88.92 ± 0.61 | 70.37 ± 0.16 | 65.50 ± 1.86 | 52.14 ± 1.25 | 79.78 ± 0.26 | 77.05 ± 0.06 |
| NODE-MoE | **89.38 ± 1.26*** | **77.78 ± 1.36** | 89.58 ± 0.60 | **73.19 ± 0.22*** | **73.64 ± 1.80*** | **62.31 ± 1.98*** | 85.37 ± 0.31 | **82.94 ± 0.06*** |

**NODE-MoE settings.** The proposed NODE-MoE framework is highly flexible, supporting a variety of gating and expert model choices. The pseudocode is provided in Appendix C.3. In this work, we use GIN (Xu et al., 2018) as the gating model. For expert models, we adopt ChebNetII (He et al., 2022) for its efficient filter learning, experimenting with 2, 3, and 5 experts initialized with different filters. More details and parameter settings are in Appendix C.4. We also experiment with using GPR-GNN as the expert backbone, and MLP and GCN as the gating models. The results can be found in Appendix D.1 and Appendix D.2, respectively.

### 4.2 PERFORMANCE COMPARISON

In this section, we evaluate NODE-MoE on both homophilic and heterophilic datasets. Node classification results are shown in Table 1, leading to the following observations:

- The proposed NODE-MoE demonstrates robust performance across both homophilic and heterophilic datasets, outperforming the baselines in most cases. This indicates its effectiveness in handling diverse graph structures.

- The GNNs and methods like LP that use fixed low-pass filters generally do well on homophilic datasets but tend to underperform on heterophilic datasets. Conversely, specialized models like LinkX, designed for heterophilic graphs, do not perform as well on homophilic datasets.

- The GNNs equipped with learnable filters generally perform well on both types of datasets, as they can adapt their filters to the dataset's structural patterns. However, their performance is still not optimal. The proposed Node-MoE, which utilizes multiple ChebNetII as experts, significantly outperforms a single ChebNetII, especially on heterophilic datasets. This result validates the effectiveness of our node-wise filtering approach.

- We also compare the proposed NODE-MoE with two MoE methods, i.e., GMoE, which adapts the receptive field for each node but still applies traditional graph convolution with low-pass filters and Mowst, which selects MLP or GNN for prediction based on the confidence of GNN. We can find NODE-MoE consistently outperforms GMoE and Mowst across all datasets.

### 4.3 ANALYSIS OF NODE-MoE

In this section, we delve into an in-depth analysis of the behaviors of NODE-MoE to demonstrate its rationality and effectiveness. We aim to uncover several key aspects of how NODE-MoE operates and performs: What specific types of filters does Node-MoE learn? Are nodes appropriately assigned to these diverse filters by the gating model? Finally, which types of nodes benefit the most from the proposed NODE-MoE for different datasets? We conduct experiments on both CiteSeer and Chameleon datasets using configurations with 2 experts. The results for the Chameleon dataset are presented below. For more results and analysis, please refer to Appendix D.

Figure 4 showcases the two filters learned by NODE-MoE on the Chameleon dataset, where filter 0 functions as a low-pass filter and filter 1 as a high-pass filter. To analyze the behavior of the gating model in NODE-MoE, we split nodes into different groups based on their homophily levels. Figure 5 displays the weights assigned by the gating model to these two experts. The results reveal that nodes with lower homophily levels predominantly receive higher weights for the high-pass filter (filter 1), and as the homophily level increases, the weight for this filter correspondingly decreases.

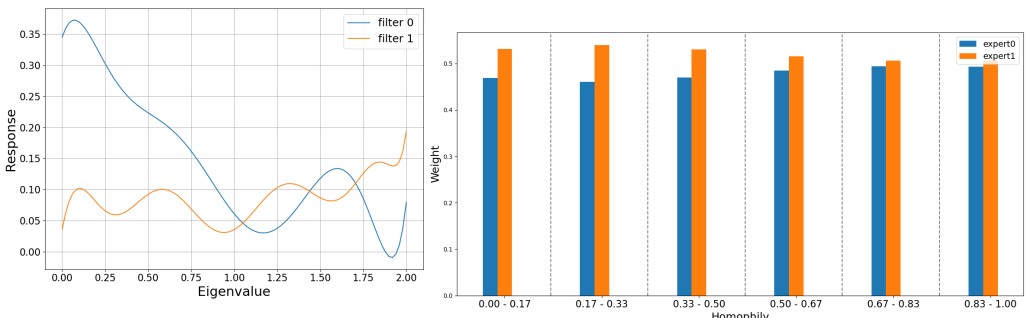

Figure 4: Learned 2 filters by NODE-MOE on Chameleon.

Figure 5: The average weight generated by the gating model for nodes in different homophily groups on Chameleon.

This pattern confirms our design that nodes with varying structural patterns require different filters, demonstrating the effectiveness of the proposed gating model. The analysis on more datasets can be found in Appendix D.3

Figure 6 presents the performance of different models on node groups with varying levels of homophily. We observe that the proposed NODE-MOE significantly improves the performance of low-homophilic nodes in the Cora dataset, and enhances the performance of high-homophilic nodes in the Chameleon dataset, compared to the single-expert ChebNetII. Moreover, NODE-MOE consistently outperforms GAT on low-homophilic nodes in both datasets. These results further validate the effectiveness of our node-wise filtering approach in adapting to diverse local structural patterns.

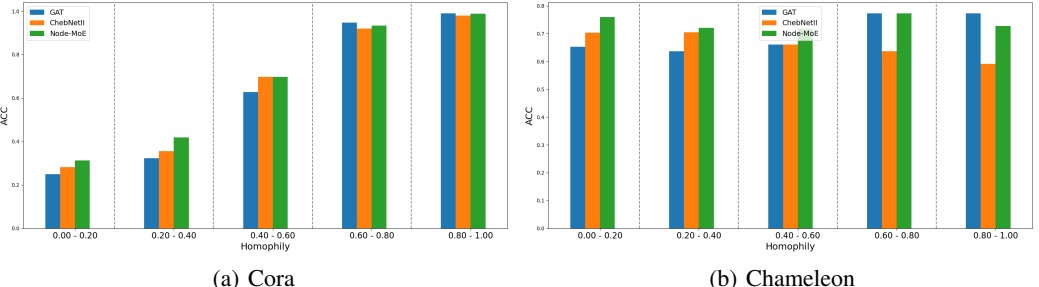

(a) Cora

(b) Chameleon

Figure 6: The performance of different models on node groups with different homophily.

### 4.4 ABLATION STUDIES

In this section, we conduct ablation studies to investigate the effectiveness of key components in the NODE-MOE framework. For the gating mechanism, we compare a standard MLP-based gating model that uses input features $\mathbf{X}$ and the Top-K gating strategy introduced in Section 3.4. As shown in Figure 7, we make two key observations: (1) the MLP-based gating performs comparably to a single ChebNetII expert and lags behind the proposed gating method in NODE-MOE; (2) Top-1 gating achieves similar performance to soft gating with all experts, demonstrating that NODE-MOE can maintain strong performance while preserving computational efficiency.

We also compared the average training time of the proposed NODE-MOE with Top-1 gating. Specifically, we select two large datasets, ogbn-arxiv and pokec, and compared the average training time of NODE-MOE with 3 and 5 experts, denoted as NODE-MOE-3 and NODE-MOE-5, respectively. As shown in Table 2, despite utilizing 3 or 5 experts, NODE-MOE 's training time remains comparable to that of the single-expert ChebNetII as the gating model only select Top-1 expert, demonstrating its efficiency.

Table 2: Average training time (s) per epoch.

| Dataset | ChebNetII | NODE-MOE-3 | NODE-MOE-5 |
|---|---|---|---|
| ogbn-arxiv | 1.57 | 1.77 | 1.93 |
| pokec | 15.58 | 16.6 | 16.79 |

Additionally, we explore the effects of the number of experts and the value of K in Top-K gating. The results, shown in the Appendix D.4 and D.5, demonstrate that NODE-MOE achieves excellent performance with just a few experts (e.g., 2) and small K values (e.g., 1). Furthermore, we explored using smaller hidden dimensions in NODE-MOE to match the number of learnable parameters in the baselines. The results in Appendix D.6 confirm that Node-MoE maintains similar performance while consistently outperforming the baselines.

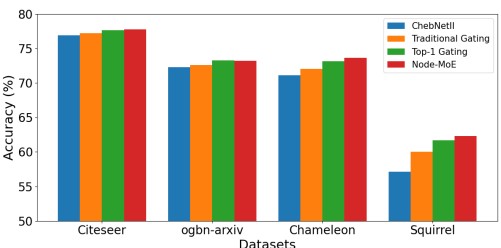
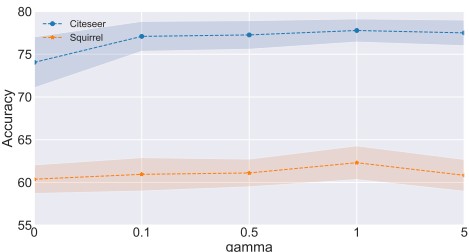

Figure 7: The performance comparison of different gating variants.

Figure 8: The performance with different weight parameters $\gamma$ of the filter smoothing loss.

We also investigate the impact of the weight parameter, $\gamma$, of the filter smoothing loss on the overall performance. As shown in Figure 8, incorporating the filter smoothing loss generally enhances performance, especially for the Citeseer dataset. For more detailed insights into the effects of the filter smoothing loss, please refer to Appendix B. Additionally, We evaluate NODE-MOE's performance under noisy feature settings. As shown in Appendix D.7, NODE-MOE still outperforms the single-expert model even at higher noise levels. Furthermore, we explore the use of diverse expert models in NODE-MOE, such as GCN and LinkX. The results shown in Appendix D.11 demonstrate the flexibility and effectiveness of NODE-MOE using different types of experts.

## 5 RELATED WORKS

Graph Neural Networks (GNNs) have been widely used across various tasks, most of which adopt the message-passing framework(Gilmer et al., 2017) and exhibit low-pass filtering behavior (Nt & Maehara, 2019; Zhao & Akoglu, 2019). While effective on homophilic graphs, these models struggle on heterophilic graphs, prompting the development of specialized models such as GloGNN (Li et al., 2022), LinkX (Lim et al., 2021a), and MixHop (Abu-El-Haija et al., 2019). Additionally, models like BernNet (He et al., 2021), GPRGNN (Chien et al., 2020), and ChebNetII (He et al., 2022) introduce learnable filters that adapt to varying structural patterns. To better capture mixed structural patterns in real-world graphs (Suresh et al., 2021; Li et al., 2022; Mao et al., 2024), several methods introduce multiple filters, including ACM-GNN (Luan et al., 2022), AutoGCN (Wu et al., 2022), PC-Conv (Li et al., 2024), and ASGAT (Li et al., 2021). Our method is distinct from these methods: they typically adopt a post-fusion strategy in which all nodes are processed by all filters, leading to increased computational cost. In contrast, NODE-MOE employs a Top-K gating mechanism that activates only the most relevant filters for each node, substantially improving efficiency. Moreover, as illustrated in Figure 1, post-fusion methods may struggle to reflect filter relevance accurately, as they assign weights solely based on the filtered node representations. By contrast, the gating mechanism in NODE-MOE is theoretically grounded in Theorem 3.

Mixture-of-Experts (MoE) architectures (Jacobs et al., 1991; Jordan & Jacobs, 1994) have been widely used to improve the efficiency of large models in NLP (Du et al., 2022; Zhou et al., 2022) and vision (Riquelme et al., 2021). In the graph domain, GraphMETRO (Wu et al., 2023) and GMoE (Wang et al., 2024) apply MoE to address distribution shifts and adapt propagation depth. Link-MoE (Ma et al., 2024) selects from different models based on node-pair heuristics for link prediction. Mowst (Zeng et al., 2023) switches between MLP and GNN predictions based on model confidence. While effective in specific settings, these methods rely on post-fusion or fixed filters, limiting their flexibility. In contrast, NODE-MOE introduces learnable, node-wise expert selection, achieving both higher adaptability and efficiency.

## 6 CONCLUSION

In this paper, we explored the complex structural patterns inherent in real-world graph datasets, which typically exhibit a mixture of homophilic and heterophilic patterns. Notably, these patterns exhibit significant variability across different communities within the same dataset, highlighting the intricate and diverse nature of graph structures. Our theoretical analysis reveals that the conventional single global filter, commonly used in many GNNs, is often inadequate for capturing such complex structural patterns. To address this limitation, we proposed the node-wise filtering method, NODE-MOE, a flexible and effective solution that adaptively selects appropriate filters for different nodes. Extensive experiments demonstrate the proposed NODE-MOE demonstrated excellent performance on both homophilic and heterophilic datasets. Further, our behavioral analysis and ablation studies validate the design and effectiveness of the proposed NODE-MOE.

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

## A   PROOFS OF THEOREMS

In this section, we present the proofs of the theorems from Section 2. These theorems analyze the separability when different filters are applied to graphs generated by a mixed CSBM model in Defination 1 - $CSBM(n, \boldsymbol{\mu}, \boldsymbol{\nu}, (p_0, q_0), (p_1, q_1), P)$ using a linear classifier.

Notably, the following proof is derived based on Baranwal et al. (2021), which analyzes the linear separability of a single graph convolution under a single CSBM model with only one pattern - $CSBM(n, \boldsymbol{\mu}, \boldsymbol{\nu}, (p, q))$. We extend the analysis to graphs with mixed CSBM models. Besides, we analyze the scenarios in which different filters are applied to the same graph.

We follow the assumption 1 and 2 in Baranwal et al. (2021): The graph size n should be relatively large with $\omega(d \log d) \le n \le O(\text{poly}(d))$, and the graph is not too sparse with $p_0, q_0, p_1, q_1 = \omega\left(\log^2(n)/n\right)$.

### A.1   PROOF OF THEOREM 1

*Proof.* For the low-pass filter, consider the filtered feature $\tilde{\mathbf{X}} = \mathbf{D}^{-1}\mathbf{A}\mathbf{X}$. Due to the normal distribution of node feature $\mathbf{X}$, the filtered feature of node $i$ still follows the normal distribution. Specifically, the mean of nodes in different classes and partterns can be represented by:

$$
m(i) = E(\tilde{\mathbf{X}}_i) = \begin{cases} \dfrac{p_0 \boldsymbol{\mu} + q_0 \boldsymbol{\nu}}{p_0 + q_0}(1 + o(1)) & \text{for } i \in C_0 \text{ and } i \in H_0 \\[2mm] \dfrac{q_0 \boldsymbol{\mu} + p_0 \boldsymbol{\nu}}{p_0 + q_0}(1 + o(1)) & \text{for } i \in C_1 \text{ and } i \in H_0 \\[2mm] \dfrac{p_1 \boldsymbol{\mu} + q_1 \boldsymbol{\nu}}{p_1 + q_1}(1 + o(1)) & \text{for } i \in C_0 \text{ and } i \in H_1 \\[2mm] \dfrac{q_1 \boldsymbol{\mu} + p_1 \boldsymbol{\nu}}{p_1 + q_1}(1 + o(1)) & \text{for } i \in C_1 \text{ and } i \in H_1 \end{cases},
$$

where $C_0$ and $C_1$ represent the class 0 and class 1, respectively; $H_0$ and $H_1$ are the homophilic and heterophilic node sets, respectively. The covariance matrix can be represented by: $Cov(\tilde{\mathbf{X}}_i) = \frac{1}{d\mathbf{D}_{ii}}\mathbf{I}$. Lemma 2 in Baranwal et al. (2021) demostrate that for any unit vector $\mathbf{w}$, we have: $\left|\left(\tilde{\mathbf{X}}_i - m(i)\right) \cdot \mathbf{w}\right| = O\left(\sqrt{\frac{\log n}{dn(p_0 + q_0)}}\right)$.

If we only consider the nodes with homophilic patterns, i.e., $i \in H_0$, we can find the optimal linear classifier with $\mathbf{w}^* = R\frac{\boldsymbol{\nu} - \boldsymbol{\mu}}{\|\boldsymbol{\nu} - \boldsymbol{\mu}\|}$ and $\mathbf{b}^* = -\frac{1}{2}\langle \boldsymbol{\nu} + \boldsymbol{\mu}, \mathbf{w}^* \rangle$. We also have the assumption that the distance between $\boldsymbol{\mu}$ and $\boldsymbol{\nu}$ are relatively large, with $\|\boldsymbol{\nu} - \boldsymbol{\mu}\| = \Omega\left(\frac{\log n}{dn(p_0 + q_0)}\right)$.

Then, for $i \in C_0$ and $i \in H_0$, we have:

$$\langle \tilde{\mathbf{X}}_i, \mathbf{w}^* \rangle + b^* = \frac{\langle p_0 \boldsymbol{\mu} + q_0 \boldsymbol{\nu}, \mathbf{w}^* \rangle}{p_0 + q_0} (1 + o(1)) +$$

$$O\left( \|\mathbf{w}^*\| \sqrt{\frac{\log n}{dn(p+q)}} \right) - \frac{1}{2} \langle \boldsymbol{\nu} + \boldsymbol{\mu}, \mathbf{w}^* \rangle$$

$$= \frac{\langle 2p_0 \boldsymbol{\mu} + 2q_o \boldsymbol{\nu} - (p_0 + q_0)(\boldsymbol{\mu} + \boldsymbol{\nu}), \mathbf{w}^* \rangle}{p_0 + q_0} (1 + o(1))$$

$$+ o(\|\mathbf{w}^*\|)$$

$$= \frac{p_0 - q_0}{2(p_0 + q_0)} \langle \boldsymbol{\mu} - \boldsymbol{\nu}, \mathbf{w}* \rangle (1 + o(1)) + o(\|\mathbf{w}^*\|)$$

$$= -\frac{R(p_0 - q_0)}{2(p_0 + q_0)} \|\boldsymbol{\mu} - \boldsymbol{\nu}\| (1 + o(1)) < 0$$

Similarly, for $i \in C_1$ and $i \in H_0$, we have:

$$\langle \tilde{\mathbf{X}}_i, \mathbf{w}^* \rangle + b^* = -\frac{R(q_0 - p_0)}{2(p_0 + q_0)} \|\boldsymbol{\mu} - \boldsymbol{\nu}\| (1 + o(1)) > 0$$

Therefore, the linear classifier with $w^*$ and $b^*$ can separate class $C_0$.

However, if we apply this linear classifier to the heterophilic node set $H_1$, where $p_1 < q_1$, we have:

$$\langle \tilde{\mathbf{X}}_i, \mathbf{w}^* \rangle + b^* = \begin{cases} -\dfrac{R(p_1 - q_1)}{2(p_1 + q_1)} \|\boldsymbol{\mu} - \boldsymbol{\nu}\| (1 + o(1)) > 0 \\ \qquad\qquad \text{for } i \in C_0 \text{ and } i \in H_1 \\ -\dfrac{R(q_1 - p_1)}{2(p_1 + q_1)} \|\boldsymbol{\mu} - \boldsymbol{\nu}\| (1 + o(1)) < 0 \\ \qquad\qquad \text{for } i \in C_1 \text{ and } i \in H_1 \end{cases}$$

Therefore, all nodes in $H_1$ are misclassified. The binary cross-entropy over node set $H_1$ can be represented by:

$$L(H_1, \mathbf{w}^*, b^*) = \frac{1}{|H_1|} \sum_{i \in H_1} -y_i \log\left( \sigma\left( \left\langle \tilde{\mathbf{X}}_i, \mathbf{w}^* \right\rangle + \tilde{b} \right) \right) -$$

$$(1 - y_i) \log\left( 1 - \sigma\left( \left\langle \tilde{\mathbf{X}}_i, \mathbf{w}^* \right\rangle + b^* \right) \right)$$

$$= \frac{1}{|H_1|} \sum_{i \in H_1} \log\left( 1 + \exp\left( (1 - 2y_i) \left( \left\langle \tilde{X}_i, \tilde{\mathbf{w}} \right\rangle + b^* \right) \right) \right)$$

$$= \log\left( 1 + \exp\left( -\frac{R(p_1 - q_1)}{2(p_1 + q_1)} \|\boldsymbol{\mu} - \boldsymbol{\nu}\| (1 + o(1)) \right) \right)$$

As for $x = -\frac{R(p_1 - q_1)}{2(p_1 + q_1)} \|\boldsymbol{\mu} - \boldsymbol{\nu}\| > 0$, we have $e^x \geq x$. As a result, we have

$$L(H_1, \mathbf{w}^*, b^*) \geq \frac{R(q_1 - p_1)}{2(p_1 + q_1)} \|\boldsymbol{\mu} - \boldsymbol{\nu}\| (1 + o(1))$$

$\square$

## A.2 Proof of Theorem 2

*Proof.* Suppose we apply a high-pass filter to the heterophilic nodes $H_1$ and the filtered features are $\tilde{\mathbf{X}} = -\mathbf{D}^{-1}\mathbf{A}\mathbf{X}$. For nodes in $H_1$,

$$m(i) = E(\tilde{\mathbf{X}}_i) = \begin{cases} -\dfrac{p_1\boldsymbol{\mu} + q_1\boldsymbol{\nu}}{p_1 + q_1}(1 + o(1)) & \text{for } i \in C_0 \text{ and } i \in H_1 \\ -\dfrac{q_1\boldsymbol{\mu} + p_1\boldsymbol{\nu}}{p_1 + q_1}(1 + o(1)) & \text{for } i \in C_1 \text{ and } i \in H_1 \end{cases}$$

Therefore, if we apply the same linear classifier with $\mathbf{w}^*$ and $b^*$, then we have:

$$\langle \tilde{\mathbf{X}}_i, \mathbf{w}^* \rangle + b^* = \begin{cases} \dfrac{R(p_1 - q_1)}{2(p_1 + q_1)}\|\boldsymbol{\mu} - \boldsymbol{\nu}\|(1 + o(1)) < 0 \\ \qquad\qquad \text{for } i \in C_0 \text{ and } i \in H_1 \\ \dfrac{R(q_1 - p_1)}{2(p_1 + q_1)}\|\boldsymbol{\mu} - \boldsymbol{\nu}\|(1 + o(1)) > 0 \\ \qquad\qquad \text{for } i \in C_1 \text{ and } i \in H_1 \end{cases}$$

As a result, the same linear classifier can separate both the homophilic set $H_0$ and heterophilic set $H_1$.

$\square$

### A.3 Proof of Theorem 3

*Proof.* Let $\mathbf{A}$ be the adjacency matrix of $G$, $\mathbf{D}$ be the diagonal degree matrix where $D_{ii}$ is the degree of node $i$, and $\mathbf{X} \in \mathbb{R}^{n \times d}$ be the feature matrix with $\mathbf{X}_i$ denoting the feature vector of node $i$. The filtered feature is defined as:

$$\tilde{\mathbf{X}} = \mathbf{D}^{-1}\mathbf{A}\mathbf{X},$$

where $\mathbf{D}^{-1}\mathbf{A}$ averages features across neighbors of each node.

We now analyze the squared feature change, $f_i^2 = (\tilde{\mathbf{X}}_i - \mathbf{X}_i)^2$, which represents the squared deviation of the aggregated feature from the original feature. For node $i$, this is:

$$f_i^2 = \left( \sum_{j \in \mathcal{N}(i)} \frac{\mathbf{X}_j}{D_{ii}} - \mathbf{X}_i \right)^2,$$

where $\mathcal{N}(i)$ is the neighborhood of $i$.

Nodes are divided into:

- $H_0$: Homophilic nodes where intra-class connections dominate.

- $H_1$: Heterophilic nodes where inter-class connections dominate.

Each node belongs to one of two classes $C_0$ or $C_1$, with the class means $\boldsymbol{\mu}$ and $\boldsymbol{\nu}$, respectively.

If $f_i = \tilde{\mathbf{X}}_i - \mathbf{X}_i \sim N(\mu_{f_i}, \sigma_{f_i}^2)$, then $f_i^2$ follows a scaled Chi-squared distribution:

$$f_i^2 \sim \frac{\sigma_{f_i}^2}{\sigma^2} \chi^2(1, \lambda_i),$$

where:

- $\chi^2(1, \lambda_i)$ is a non-central Chi-squared distribution with 1 degree of freedom and non-centrality parameter $\lambda_i = \frac{\mu_{f_i}^2}{\sigma_{f_i}^2}$.

- $\sigma_{f_i}^2 = \frac{\sigma^2}{d}\left(1 + \frac{1}{D_{ii}}\right)$ for node $i$.

- The mean $\mu_{f_i}$ depends on the node type (homophilic or heterophilic):

$$\mu_{f_i} = \begin{cases} \frac{q_0}{p_0+q_0}(\nu - \mu), & \text{if } i \in H_0, C_0, \\ -\frac{q_0}{p_0+q_0}(\nu - \mu), & \text{if } i \in H_0, C_1, \\ \frac{q_1}{p_1+q_1}(\nu - \mu), & \text{if } i \in H_1, C_0. \\ -\frac{q_1}{p_1+q_1}(\nu - \mu), & \text{if } i \in H_1, C_1. \end{cases}$$

For each node $i$, the expected squared feature change $f_i^2$ is:

$$\mathbb{E}[f_i^2] = \mu_{f_i}^2 + \sigma_{f_i}^2,$$

and the variance of $f_i^2$ is:

$$\text{Var}(f_i^2) = 2\sigma_{f_i}^4 + 4\mu_{f_i}^2 \sigma_{f_i}^2.$$

Misclassification occurs when the squared feature changes of nodes $i \in H_0$ and $j \in H_1$ overlap. Define the difference in squared feature changes:

$$DF_{ij} = f_i^2 - f_j^2.$$

The expectation of $DF_{ij}$ is:

$$\mathbb{E}[DF_{ij}] = \mathbb{E}[f_i^2] - \mathbb{E}[f_j^2].$$

Substituting $\mathbb{E}[f_i^2] = \mu_{f_i}^2 + \sigma_{f_i}^2$, we get:

$$\mathbb{E}[DF_{ij}] = (\mu_{f_i}^2 - \mu_{f_j}^2) + (\sigma_{f_i}^2 - \sigma_{f_j}^2),$$

where:

- The difference in means $\mu_{f_i}^2 - \mu_{f_j}^2$ is:

$$\mu_{f_i}^2 - \mu_{f_j}^2 = \left( \frac{q_0}{p_0 + q_0} - \frac{q_1}{p_1 + q_1} \right)^2 (\nu - \mu)^2 = \Delta^2 (\nu - \mu)^2.$$

Here, $\Delta = \frac{q_0}{p_0+q_0} - \frac{q_1}{p_1+q_1}$ represents the normalized connection bias between classes.

- The variance difference $\sigma_{f_i}^2 - \sigma_{f_j}^2$ is:

$$\sigma_{f_i}^2 - \sigma_{f_j}^2 = \frac{\sigma^2}{d} \left( \frac{1}{D_{ii}} - \frac{1}{D_{jj}} \right).$$

As $d \to \infty$, this term vanishes.

Thus:

$$\mathbb{E}[DF_{ij}] = \Delta^2 (\nu - \mu)^2 + \mathcal{O}\left(\frac{1}{d}\right).$$

The variance of $DF_{ij}$ is:

$$\text{Var}(DF_{ij}) = \text{Var}(f_i^2) + \text{Var}(f_j^2).$$

For each node:

$$\text{Var}(f_i^2) = 2\sigma_{f_i}^4 + 4\mu_{f_i}^2 \sigma_{f_i}^2.$$

Since $\sigma_{f_i}^2 = \frac{\sigma^2}{d}\left(1 + \frac{1}{D_{ii}}\right)$ and $\mu_{f_i}^2 \propto \frac{1}{d}$, we get:

$$\text{Var}(f_i^2) \sim \mathcal{O}\left(\frac{1}{d}\right).$$

The misclassification probability $\mathbb{P}(DF_{ij} \leq \epsilon)$ can be bounded using the Chernoff inequality:

$$\mathbb{P}(DF_{ij} \leq \epsilon) \leq \exp\left( -\frac{(\mathbb{E}[DF_{ij}] - \epsilon)^2}{2\,\text{Var}(DF_{ij})} \right).$$

Substituting the results:

$$\mathbb{E}[DF_{ij}] = \Delta^2(\nu - \mu)^2 + \mathcal{O}\left(\frac{1}{d}\right), \quad \text{Var}(DF_{ij}) \sim \mathcal{O}\left(\frac{1}{d}\right).$$

Thus:

$$\frac{(\mathbb{E}[DF_{ij}] - \epsilon)^2}{2\,\text{Var}(DF_{ij})} \sim \mathcal{O}(d),$$

implying that the exponential decay in the Chernoff bound becomes increasingly sharp as $d \to \infty$, making $\mathbb{P}(DF_{ij} \leq \epsilon)$ approach 1.

$\square$

## B   THE IMPACT OF FILTER SMOOTHING LOSS

In this section, we explore the impact of the proposed filter smoothing loss on the behavior of the learned filters in our NODE-MOE framework. Figures 9 and 10 display the effects of the NODE-MOE framework without and with the application of filter smoothing loss, respectively. Without the filter smoothing loss, as shown in Figure 9, the learned filters exhibit significant oscillations, making it challenging to discern their specific functions. In contrast, with the filter smoothing loss applied, as illustrated in Figure 10, the behavior of the filters becomes more distinct: filter 0 clearly functions as a low-pass filter, and filter 1 as a high-pass filter.

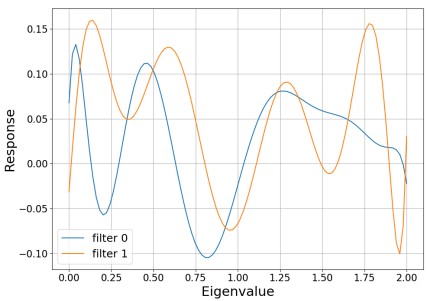
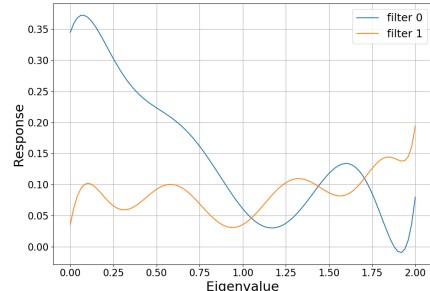

Figure 9: Learned 2 filters by NODE-MOE on Chameleon without filter smoothing loss.

Figure 10: Learned 2 filters by NODE-MOE on Chameleon with filter smoothing loss.

Additionally, we assessed the training dynamics of the proposed Node-MoE framework by comparing performance with and without the filter smoothing loss, while keeping other hyperparameters constant. For the Citeseer dataset, applying the filter smoothing loss resulted in a higher average training accuracy of 99.37 ± 0.17, compared to 93.51 ± 1.27 when the loss was not applied. A similar pattern was observed on the Squirrel dataset, where the training accuracy was 96.54 ± 1.42 with the filter smoothing loss, versus 95.54 ± 0.94 without it. These results suggest that oscillations in the filters without the smoothing loss can hinder the model's ability to fit the data effectively, resulting in suboptimal performance as shown in Section 4.4.

## C   DATASETS AND EXPERIMENTAL SETTINGS

In this section, we detail the datasets used and the experimental settings for both the baseline models and the proposed NODE-MOE framework.

### C.1   DATASETS

We conduct experiments across seven widely recognized datasets, which encompass both homophilic and heterophilic types. The homophilic datasets include Cora, CiteSeer, and Pubmed (Sen et al., 2008), along with ogbn-arxiv (Hu et al., 2020); the heterophilic datasets comprise Chameleon, Squirrel (Pei et al., 2020), Penn94 and pokec Lim et al. (2021a). For Cora, CiteSeer, and Pubmed, we

generate ten random splits, allocating nodes into training, validation, and testing sets with proportions of 60%, 20%, and 20%, respectively. For the heterophilic datasets, we adhere to the ten fixed splits as defined in Pei et al. (2020). The ogbn-arxiv dataset is assessed using its standard split as established by (Hu et al., 2020). Detailed statistics of these datasets are shown in Table 3.

Table 3: Statistics of datasets. The split ratio is for train/validation/test.

|  | Homophilic Datasets | | | | Heterophilc Datasets | | | |
|  | Cora | CiteSeer | PubMed | ogbn-arxiv | Chameleon | Squirrel | Penn94 | pokec |
|---|---|---|---|---|---|---|---|---|
| #Nodes | 2,708 | 3,327 | 19,717 | 169, 343 | 2,277 | 5,201 | 41,554 | 1,632,803 |
| #Edges | 5,429 | 4,732 | 44,338 | 1, 166, 243 | 31,421 | 198,493 | 1,362,229 | 30,622,564 |
| #Classes | 7 | 6 | 3 | 40 | 5 | 5 | 2 | 2 |
| #Node Features | 1,433 | 3,703 | 500 | 128 | 2,325 | 2,089 | 4814 | 65 |
| #Split Ratio | 60/20/20 | 60/20/20 | 60/20/20 | 54/18/28 | 48/32/20 | 48/32/20 | 50/25/25 | 50/25/25 |

## C.2 RESULTS ON ADDITIONAL HETEROPHILIC DATASETS FROM PLATONOV ET AL. (2023)

Platonov et al. Platonov et al. (2023) introduced a set of heterophilic graphs with diverse properties and showed that standard GNNs often outperform specialized models on these datasets. We further evaluate the proposed NODE-MOE on two of their benchmarks: amazon-ratings and tolokers. Specifically, we report Accuracy on amazon-ratings and ROC AUC on tolokers, following the settings in Platonov et al. (2023). The results are presented in Table 4, where NODE-MOE achieves the best performance on both datasets, further validating its effectiveness on newly introduced heterophilic benchmarks.

Table 4: The results on amazon-ratings and tolokers datasets.

|  | amazon-ratings | tolokers |
|---|---|---|
| GCN Kipf & Welling (2016) | 48.70 ± 0.63 | 83.64 ± 0.67 |
| SAGE Hamilton et al. (2017) | 53.63 ± 0.39 | 82.43 ± 0.44 |
| GAT Veličković et al. (2017) | 49.09 ± 0.63 | 83.70 ± 0.47 |
| GAT-sep Platonov et al. (2023) | 52.70 ± 0.62 | 83.78 ± 0.43 |
| Graph Transformer(GT) Shi et al. (2020) | 51.17 ± 0.66 | 83.23 ± 0.64 |
| GT-sep Platonov et al. (2023) | 52.18 ± 0.80 | 82.52 ± 0.92 |
| GPR-GNN Chien et al. (2020) | 44.88 ± 0.34 | 72.94 ± 0.97 |
| ChebNetII He et al. (2022) | 52.23 ± 0.58 | 82.52 ± 0.98 |
| Node-MoE | **54.02 ± 0.71** | **84.87 ± 0.92** |

## C.3 ALGORITHM OF NODE-MOE

The algorithm of NODE-MOE is shown in Algorithm 1. Lines 2-3 initialize the filters of experts based on the setting in Section C.4. Line 4 calculates the input for the gating model. Lines 6-7 calculate the prediction with top-k gating. Line 8 update the model based on the loss in Section 3.3.

---

**Algorithm 1:** NODE-MOE

---

1 Input graph $\mathbf{A}$, Node feature $\mathbf{X}$, $m$ experts, i.e., $E_1, E_2, \ldots, E_m$, Gating model $g$, Top-K gating $k$

2 **for** $i = 1, 2, \ldots, m$ **do**

3 $\quad$ Initialize the filter $i$-th expert $E_i$

4 Calculate the gating input $\mathbf{GX} = [\mathbf{X}, |\mathbf{AX} - \mathbf{X}|, |\mathbf{A}^2\mathbf{X} - \mathbf{X}|]$

5 **repeat**

6 $\quad$ $\mathbf{G} = \text{Softmax}(\text{KeepTopK}(g(\mathbf{GX}), k))$

7 $\quad$ $\hat{y} = \sum_{o=0}^{m} \mathbf{G}_o E_o(\mathbf{A}, \mathbf{X})$

8 $\quad$ Update NODE-MOE weight by gradient descent on L

9 **until** *Model converges*;

---

## C.4 Experimental Settings

For the baseline models, we adopt the same parameter setting in their original paper. For the proposed NODE-MOE, we adopt GCNII as the experts. Specifically, for smaller datasets, we use GIN as the gating model, while for larger datasets, such as Pokec, we use an MLP as the gating model. Notably, the GCNII model has different learning rates and weight decay for the filters and other parameters. All the hyperparameters are tuned based on the validation accuracy from the following search space:

- Gating Learning Rate: {0.0001, 0.001, 0.01 }
- Gating Dropout: {0, 0.5, 0.8}
- Gating Weight Decay: {0, 5e-5, 5e-4}
- Expert Learning Rate for Filters: {0.001, 0.01, 0.1}
- Expert Weight Decay for Filters: {0, 5e-5, 5e-3, 5e-2 }
- Expert Learning Rate: {0.001, 0.01, 0.1, 0.5}
- Expert Dropout: {0, 0.5, 0.8}
- Filter Smoothing loss weight: {0, 0.01, 0.1, 1}
- Load balancing weight for top-k gating: {0, 0.001, 0.01, 0.1, 1}
- Number of experts: {2, 3, 5}

For the initialization of filters in ChebNetII, which uses a K-order approximation, we employ a set of initialization strategies for the polynomial coefficients. These strategies include: decreasing powers $[\alpha^0, \alpha^1, \cdots, \alpha^K]$, increasing powers $[\alpha^K, \alpha^{K-1}, \cdots, \alpha^0]$, and uniform values $[1, 1, \cdots, 1]$. For configurations with 2 or 3 experts, we set $\alpha = 0.9$. When expanding to 5 experts, we use two values of $\alpha$, setting them at $0.9$ and $0.8$, respectively, to diversify the response characteristics of the filters. The code of the proposed NODE-MOE can be found via: https://anonymous.4open.science/r/Node-MoE-A05D/.

We use a single GPU of NVIDIA RTX A5000 24Gb, to run the experiments.

## D  ANALYSIS OF THE PROPOSED NODE-MOE

In this section, we provide more analysis of the proposed NODE-MOE by comprehensive experiments.

### D.1  NODE-MOE WITH GPRGNN AS EXPERT BACKBONE

In the Section 4, we selected ChebNetII as the expert backbone. In this section, we experiment with GPR-GNN as the expert model. The results in Table 5 show that Node-MoE with GPR-GNN significantly enhances the performance of the base GPR-GNN model, further demonstrating the flexibility and effectiveness of our approach.

Table 5: NODE-MOE with different expert backbones.

| Method | Cora | CiteSeer | Chameleon | Squirrel |
|---|---|---|---|---|
| GPR-GNN | 88.54 ± 0.67 | 76.44 ± 1.89 | 62.85 ± 2.90 | 54.35 ± 0.87 |
| Node-MoE (GPR-GNN) | 89.30 ± 1.44 | 77.12 ± 1.60 | 70.88 ± 2.22 | 56.92 ± 0.61 |
| ChebNetII | 88.71 ± 0.93 | 76.93 ± 1.57 | 71.14 ± 2.13 | 57.12 ± 1.13 |
| Node-MoE (ChebNetII) | 89.38 ± 1.26 | 77.78 ± 1.36 | 73.64 ± 1.80 | 62.31 ± 1.98 |

### D.2  NODE-MOE WITH DIFFERENT MODELS AS GATING BACKBONE

In Section 4, we selected GIN as the gating model backbone. In this section, we experiment with alternative gating models, including GCN and MLP. The results in Table 6 show that GNN-based models, such as GCN and GIN, generally outperform MLP, particularly on heterophilic datasets. These findings align with the analysis in Section 2, which highlights the significant variation in homophily across different communities and supports the rationale for using a gating model.

Table 6: NODE-MOE with different gating backbones.

| Gating Model | Cora | CiteSeer | Chameleon | Squirrel |
|---|---|---|---|---|
| MLP | 89.26 ± 1.34 | 77.59 ± 1.27 | 72.70 ± 1.75 | 59.88 ± 2.33 |
| GCN | 89.28 ± 0.96 | 77.89 ± 1.29 | 73.29 ± 1.71 | 62.04 ± 1.92 |
| GIN | 89.38 ± 1.26 | 77.78 ± 1.36 | 73.64 ± 1.80 | 62.31 ± 1.98 |

### D.3 THE BEHAVIOR OF NODE-MOE WITH 2 EXPERTS

The learned filters and the corresponding gating weights for nodes with different homophily levels are illustrated below. For the Chameleon dataset, these are displayed in Figure 11 for the filters and Figure 12 for the gating weights. Similarly, for the Citeseer dataset, the filters are shown in Figure 13 and the gating weights in Figure 14.

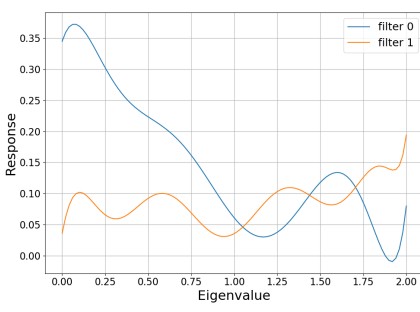

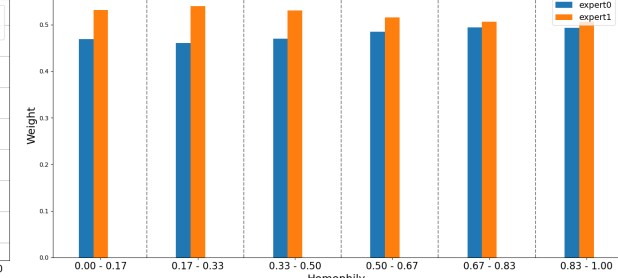

Figure 11: Learned 2 filters by NODE-MOE on Chameleon.

Figure 12: The average weight generated by the gating model for nodes in different homophily groups on Chameleon.

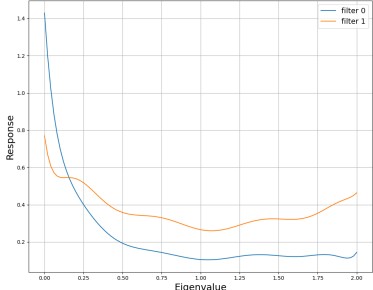

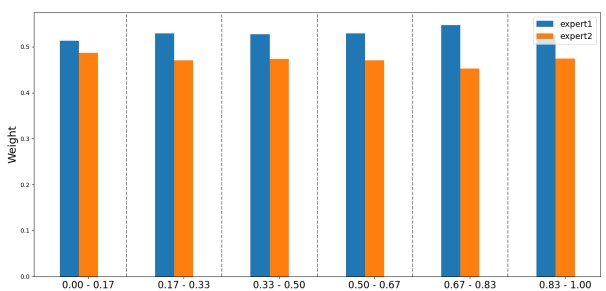

Figure 13: Learned 2 filters by NODE-MOE on Citeseer.

Figure 14: The average weight generated by the gating model for nodes in different homophily groups on Citeseer.

For both datasets, the learned filters demonstrate distinct characteristics: filter 0s function as low-pass filters, effectively smoothing signals, while filter 1s respond more strongly to high-frequency signals, characteristic of high-pass filters. Specifically, for the heterophilic dataset, such as Chameleon, the gating model generally assigns higher weights to filter 1, indicating a preference for high-pass filtering to accommodate the less homophilic nature of the dataset. Conversely, for the homophilic dataset, such as Citeseer, higher weights are typically assigned to filter 0, emphasizing low-pass filtering.

Moreover, within the Chameleon dataset, the weight assigned to the high-pass filter (filter 1) decreases as the homophily level increases. In contrast, in the Citeseer dataset, the weight to the low-pass filter (filter 0) increases with rising homophily levels. This pattern supports our initial hypothesis: nodes with lower homophily are better served by high-pass filters to capture the dissimilarity among neighbors, while nodes with higher homophily benefit from low-pass filters to reinforce the similarity among neighboring nodes.

Table 7: The performance of Node-MoE with different number of experts.

| Experts | 1 | 2 | 3 | 5 |
|---|---|---|---|---|
| Cora | 88.71 ± 0.93 | 89.19 ± 1.53 | 89.38 ± 1.26 | 89.47 ± 0.85 |
| Chameleon | 71.14 ± 2.13 | 73.55 ± 1.74 | 73.64 ± 1.80 | 73.42 ± 1.43 |

Table 8: The performance of Top-K gating for Node-MoE with 3 experts.

| K | Single Expert | 1 | 2 | 3 |
|---|---|---|---|---|
| Cora | 88.71 ± 0.93 | 89.58 ± 1.44 | 89.56 ± 1.39 | 89.38 ± 1.26 |
| Chameleon | 71.14 ± 2.13 | 73.18 ± 1.45 | 73.37 ± 1.86 | 73.64 ± 1.80 |

## D.4 EFFECT OF THE NUMBER OF EXPERTS IN NODE-MOE

In this section, we analyze the impact of using different numbers of experts in NODE-MOE with soft gating. Specifically, we experiment with 1, 2, 3, and 5 experts on the Cora and Chameleon datasets, following the same settings as outlined in Section 4.1. The experimental results, shown in Table 7, indicate that NODE-MOE achieves excellent performance with only a few experts. Notably, even with just 2 experts, it outperforms the baseline models.

## D.5 EFFECT OF THE NUMBER OF SELECTED EXPERTS IN TOP-K GATING

In this section, we explore the impact of the number of selected experts K in NODE-MOE with Top-K gating. Specifically, we use 3 experts in the MoE and vary K in [1, 2, 3] in the the Top-K gating. The results in Table 8 demonstrate that even with Top-1 gating, Node-MoE achieves superior performance, highlighting its effectiveness and maintaining good efficiency.

## D.6 NODE-MOE WITH SMALLER HIDDEN DIMENSIONS

We further conducted experiments with Node-MoE using 3 experts and a significantly reduced hidden dimension size, decreasing it from 64 to 21. This adjustment ensures that the proposed Node-MoE has a comparable number of learnable parameters to models like MLP and GCN. Despite the lower hidden dimension, the results in Table 9 demonstrate that Node-MoE maintains similar performance and continues to outperform the baselines.

## D.7 NODE-MOE WITH NOISE FEATURE SEETING

The effectiveness of the gating model in NODE-MOE depends on the quality of node features, and noisy features can hinder its ability to accurately classify node patterns. In this section, we investigate the impact of noisy features on NODE-MOE. Specifically, we add varying levels of Gaussian noise to the features in Cora and Chameleon dataset, i.e., $X = X + \epsilon \mathcal{N}(0, 1)$ with $\epsilon \in [0, 0.01, 0.03, 0.05]$, where $\mathcal{N}(0, 1)$ is the standard normal distribution.

Table 9: The performance of NODE-MOE with different hidden dimensions.

| Hidden Size | Cora | CiteSeer | Chameleon | Squirrel |
|---|---|---|---|---|
| Node-MoE 21 | 89.36 ± 1.00 | 77.93 ± 1.52 | 73.51 ± 1.77 | 60.93 ± 1.89 |
| Node-MoE 64 | 89.38 ± 1.26 | 77.78 ± 1.36 | 73.64 ± 1.80 | 62.31 ± 1.98 |

Table 10: Node classification performance with different levels of noise in the node features.

| Dataset | Cora | | | | Chameleon | | | |
|---|---|---|---|---|---|---|---|---|
| Noise | 0 | 0.01 | 0.03 | 0.05 | 0 | 0.01 | 0.03 | 0.05 |
| MLP | 76.49 ± 1.13 | 59.08 ± 2.36 | 31.05 ± 1.23 | 29.58 ± 1.39 | 48.11 ± 2.23 | 30.31 ± 1.74 | 23.71 ± 1.79 | 21.80 ± 1.76 |
| GAT | 88.68 ± 1.13 | 87.78 ± 0.98 | 85.35 ± 0.98 | 84.61 ± 1.29 | 65.29 ± 2.54 | 64.47 ± 2.77 | 63.73 ± 2.19 | 62.92 ± 2.42 |
| ChebyNetII | 88.71±0.93 | 87.90 ± 1.41 | 86.25 ± 1.62 | 85.85 ± 1.09 | 71.14 ± 2.13 | 71.45 ± 1.87 | 71.54 ± 1.48 | 71.62 ± 1.56 |
| NODE-MOE | 89.38 ± 1.26 | 87.98 ± 1.51 | 86.45 ± 1.35 | 86.05 ± 1.19 | 73.64 ± 1.80 | 73.16 ± 1.18 | 72.13 ± 1.62 | 71.95 ± 1.66 |

The results on Cora and Chameleon dataset are shown in Table 10. As the noise level increases, the performance gap between NODE-MOE and the single-expert ChebyNetII decreases. However, NODE-MOE consistently outperforms the single expert, even with higher noise levels. The reason is that when noise is too high, the gating model may randomly assign nodes to different experts, making the learned filters converge to a performance similar to the single-expert model.

Table 11: Comparison between the fixed filters and learnable filters.

| Method | Cora | CiteSeer | Chameleon | squirrel |
|---|---|---|---|---|
| NODE-MOE-Fixed | 87.26 ± 1.79 | 76.15 ± 1.99 | 71.78 ± 3.37 | 56.96 ± 1.51 |
| NODE-MOE | 89.38 ± 1.26 | 77.78 ± 1.36 | 73.64 ± 1.80 | 62.31 ± 1.98 |

Table 12: Node classification performance with filtered datasets.

| Method | Chameleon | Squirrel | Chameleon-filterd | Squirrel-filtered |
|---|---|---|---|---|
| PCNet | 41.23 ± 1.42 | 26.28 ± 0.32 | 34.51 ± 1.86 | 33.08 ± 0.20 |
| ASGAT | 66.50 ± 2.80 | 55.80 ± 3.20 | 37.4 ± 6.40 | 35.10 ± 1.30 |
| ACM-GCN | 69.62 ± 1.22 | 57.02 ± 0.79 | 37.78 ± 2.28 | 36.59 ± 1.75 |
| Mowst | 65.50 ± 1.86 | 52.14 ± 1.25 | 43.45 ± 3.90 | 38.04 ± 2.14 |
| NODE-MOE | 73.64 ± 1.80 | 62.31 ± 1.98 | 43.32 ± 3.56 | 42.37 ± 1.98 |

Table 13: Semi-supervised Node classification performance with low labeling rate.

| | Cora 20 | Chameleon 20% |
|---|---|---|
| GCN | 79.41 ± 1.30 | 56.71 ± 1.72 |
| LinkX | 52.93 ± 3.04 | 60.62 ± 1.93 |
| GMoE | 76.00 ± 1.14 | 65.18 ± 1.45 |
| ChebNetII | 81.20 ± 1.04 | 64.66 ± 1.86 |
| Node-MoE | 82.12 ± 1.19 | 68.81 ± 1.96 |

Table 14: The performance of NODE-MOE with diverse experts.

| | Cora | CiteSeer | Chameleon | Squirrel |
|---|---|---|---|---|
| LinkX | 82.89 ± 1.27 | 70.05 ± 1.88 | 68.42 ± 1.38 | 61.81 ± 1.80 |
| GCN | 88.60 ± 1.19 | 76.88 ± 1.78 | 67.96 ± 1.82 | 54.47 ± 1.17 |
| Node-MoE(LinkX+GCN) | 88.92 ± 2.00 | 77.02 ± 2.03 | 73.84 ± 1.25 | 64.69 ± 1.84 |

## D.8 EFFECT OF LEARNABLE FILTERS IN NODE-MOE

The propose NODE-MOE leverages ChebNetII as the experts, which automatically learn the filters. In contrast, a few prior works use multiple fixed filters. To evaluate the effect of learnable filters, we conducted experiments with fixed filters. Specifically, we used 3 experts in NODE-MOE and fixed the filters in each expert to predefined types (low-pass, high-pass, and all-pass), referred to as NODE-MOE-fixed.

The performance comparison between NODE-MOE and NODE-MOE-fixed is shown in Table 11. NODE-MOE with learnable filters achieves better performance than with fixed filters across all datasets. This suggests that learnable filters are better suited for capturing the complex patterns present in real-world graphs.

## D.9 DUPLICATES IN THE CHAMELEON AND SQUIRREL DATASETS

Platonov et al. Platonov et al. (2023) identified that some nodes in the Chameleon and Squirrel datasets are duplicated, sharing identical neighbors, which may affect the performance of GNNs. To further validate the effectiveness of the proposed method, we also tested it on filtered versions of the Chameleon and Squirrel datasets, where the duplicate nodes were removed, referred to as Chameleon-filtered and Squirrel-filtered.

As shown in Table 12, NODE-MOE continues to perform well on these filtered datasets, demonstrating its robustness.

### D.10   NODE-MOE WITH LOW LABELING RATE

We also evaluate the semi-supervised node classification performance under a low labeling rate. Specifically, we randomly select 20 training nodes per class for the Cora dataset and use 20% of the training nodes for the Chameleon dataset. The results are shown in Table 13. Even under these low-labeling rate conditions, the proposed NODE-MOE continues to outperform the baseline models.

### D.11   NODE-MOE WITH DIVERSE EXPERTS

We explored the use of diverse expert models by selecting GCN and LinkX as two representative architectures tailored for homophilic and heterophilic patterns, respectively. We integrated these two models as the experts in Node-MoE and conducted end-to-end training. The results are summarized in the Table 14.

GCN performs exceptionally well on homophilic datasets, while LinkX excels on heterophilic datasets. The proposed Node-MoE, utilizing both GCN and LinkX as experts, achieves better performance across both homophilic and heterophilic datasets. This demonstrates the flexibility and effectiveness of Node-MoE in leveraging diverse expert models to adapt to varying structural patterns within graphs.

