# OpenReview forum: "Node-wise Filtering in Graph Neural Networks: A Mixture of Experts Approach"
_ICLR.cc/2026/Conference — Submitted to ICLR 2026_

### Official Review · Reviewer_Jq48 · 2025-10-24

**Soundness:** 3
**Presentation:** 3
**Contribution:** 2
**Rating:** 4
**Confidence:** 4

**Summary:**

This work proposes NODE-MoE, a mixture of experts approach for GNNs that aims to select node-specific filters to help improve learning over both homophilous and heterophilous nodes in a graph. To decide which filter to use, the gating mechanism uses a set of feature similarity-style terms across a node's neighborhood, routing to either a soft mixture of all experts or a top-k set of experts. On both homophilic and heterophilic datasets, NODE-MoE demonstrates moderate performance gains across a variety of baselines. Moreover, this is supported by theoretical analysis on a CSBM model with a linear classifier that demonstrates a global filter is insufficient to learn across nodes with different patterns. Overall, I think the paper does a good job of presenting a reasonable idea, but with some shortcomings in justifying some aspects of it.

**Strengths:**

1. I believe the presentation of the work is a strong point, it is relatively easy to follow and the ideas logically build on one another throughout the work. I like the general flow of the paper.
2. The empirical results seem promising, and I appreciate the authors added larger datasets, like Pokec, to demonstrate the scalability of the method. While I did not spend a ton of time going through the appendix, I also see there are tons of additional ablations and analyses, which I think is beneficial for the work.
3. I think the theory is a good starting point to establishing why we would want to use an MoE style framework with different expert designs.

**Weaknesses:**

1. I have a few concerns about the theoretical analysis and what it conveys about the work.
    - First, the theory relies on a relatively simple set up, with a linear model and binary classification. I don't feel as though the authors necessarily justify this choice, and explain why it is okay to neglect the designs that typically go into GNNs.
    - Second, theorem 1 and 2 seems to argue the failure modes of GNNs with a global filter, but not how to solve the issue. The authors include theorem 3 to address this, explaining why it is okay to use the features to separate classes. However, I don't feel as though theorem 3 actually justifies the design. Specifically, the authors set up the CSBM in such a way that the feature-neighbor similarity correlates with homophily, and the features and structure are aligned. Thus, the theorem is just re-stating the assumptions of the CSBM model, as opposed to justifying why using these features would be meaningful in real-world applications. Why I think this is problematic is the authors argue that previous works are designed around unjustified heuristics, yet, these are also heuristics that are only justified in a very particular setting.

2. Building on the point around heuristics, the other core justification the authors make for their method is moving beyond post-fusion/fixed-filters used by previous methods. Specifically, the authors claim that the advantage of their method comes from their node-wise filtering. However, I feel like, again, this is driven by heuristics, rather than a fundamental difference to previous methods. Within Node-MoE, the authors still use soft-routing which leverages all filters, but later apply a top-K gating to decide which to keep. There is no clear justification for why this would work, and feels like a heuristic-based choice that is not sufficiently justified over previous methods.

3. The performance results are decent, but mixed in some cases. I do not think this is a true negative, assuming the follow-up analysis can thoroughly explain why certain behaviors are happening. That said, I think there are some weaknesses in the post-hoc analysis that do not necessarily convince me that Node-MoE is doing exactly what the authors claim.
    - First, in figure 5, the authors show that the average weights change across homophily level. Yet, expert 1 (the high-pass filter) receives the highest scores for all bins. If this is the case, what is the point of the other expert? I also feel the authors did not provide enough justification for figures 4 and 5. For instance, on line 391 the authors say "This pattern confirms our design that nodes with varying structural patterns require different filters, demonstrating the effectiveness of the proposed gating model", yet this is the premise -- we already know nodes require different filters. I would expect an actual takeaway to show that Node-MoE effectively exploits this information and can adequately route nodes to different models that are aligned with their needs. If we were to take my point above about the weights and the dominance of expert 1, it makes it seem like routing is not actually happening.
    - For figure 6, what is the performance of the more difficult regions, like the heterophilous nodes of Cora and homophilous nodes of Chameleon, when using an MLP? One concern I have is that the GNN models are known to under-perform a basic MLP in these regions,  and it possible that simply using an MLP in these regions can outperform Node-MoE. Moreover, I would be interested to see Mowst for Figure 6 given it routes between GNN and MLP. The more fundamental question I am asking is whether Node-MoE actually is leveraging the graph structure in these other areas, or simply choosing to rely less on it.

**Questions:**

1. Can you provide more justification as to why the strong assumptions for theorem 1 and 2 are justified? If they aren't, is it possible to relax some of these?
2. Less of a question, but I think Theorem 3 needs to be reconsidered given the points I made above.
3. I think the paper uses very strong wording around why previous methods are inferior, but then tend to fall into the same paradigms. This can potentially get solved by bolstering the theory, but I also think the paper needs re-framing to not over claim.
4. Can the authors further discuss, going beyond figures 4 and 5, what is happening with the routing and how we can be certain the MoE framework is behaving as expected?
5. For figure 6, what happens if you use an MLP or Mowst and regenerate these plots.

---

> ### Author Response · Authors · 2025-11-22
> **Response to Reviewer Jq48 - Part 1**
>
> Dear Reviewer Jq48,
>
> We appreciate your constructive feedback. We are pleased to provide detailed responses to address your concerns.
>
> > W1 & Q1: First, the theory relies on a relatively simple set up, with a linear model and binary classification. I don't feel as though the authors necessarily justify this choice, and explain why it is okay to neglect the designs that typically go into GNNs.
>
> Our theoretical analysis is intentionally grounded in the (mixed-)CSBM assumption, which models graphs containing both homophilic and heterophilic generative patterns with two classes. This formulation is standard and has been widely adopted in theoretical studies of graph filtering and the expressive limitations of GNNs [1, 2, 3], precisely because it provides an analytically tractable setting while still capturing the fundamental structural challenges that arise in real-world mixed-homophily graphs.
>
> Within this framework, Theorem 1 establishes that applying a single global filter causes the filtered node features of the two classes to collapse into the same distribution. This collapse is a property of the filtering operation itself, not of the classifier. Once the class-conditional distributions become identical, no downstream classifier—whether linear, nonlinear, shallow, or deep—can recover separability, because the information required to distinguish the classes has already been erased by the global filter. This directly motivates the need for multiple distinct filters and the proposed Node-MoE.
>
> [1] "Graph convolution for semi-supervised classification: Improved linear separability and out-of-distribution generalization", ICML'21
>
> [2] "Is homophily a necessity for graph neural networks?", ICLR'22.
>
> [3] "Demystifying structural disparity in graph neural networks: Can one size fit all?", NeurIPS'23
>
> > W2 & Q2: Second, theorem 1 and 2 seems to argue the failure modes of GNNs with a global filter, but not how to solve the issue. The authors include theorem 3 to address this, explaining why it is okay to use the features to separate classes. However, I don't feel as though theorem 3 actually justifies the design. Specifically, the authors set up the CSBM in such a way that the feature-neighbor similarity correlates with homophily, and the features and structure are aligned. Thus, the theorem is just re-stating the assumptions of the CSBM model, as opposed to justifying why using these features would be meaningful in real-world applications. Why I think this is problematic is the authors argue that previous works are designed around unjustified heuristics, yet, these are also heuristics that are only justified in a very particular setting.
>
> **A2:** We appreciate the reviewer’s concern. We first clarify the role of Theorem 3 and why it is not merely restating the assumptions of CSBM.
>
> While the mixed-CSBM assumes correlations between features, labels, and structure, these assumptions do not make feature-based separation trivial. In mixed homophily/heterophily graphs, the marginal feature distributions of different classes become highly overlapped, which is exactly why a global filter collapses class information and why no downstream classifier—linear or nonlinear—can recover separability. This is the core failure mode highlighted in our theoretical analysis. The key point of our Theorem 3 is that, even when raw features and globally filtered features lose class-level separability, the local deviation between a node’s feature and its neighbors’ aggregated features remains discriminative. This quantity captures whether a node lies in a homophilic or heterophilic region and is strictly more informative than the raw features alone. This insight directly motivates our gating design: instead of relying on post-filter embeddings (as prior methods do), we leverage the one statistic that theory guarantees remains informative under mixed homophily/heterophily.
>
> Although CSBM is a simplified model, this type of feature–structure alignment is repeatedly observed in real-world graphs. To illustrate, we directly apply the 1-hop feature distance (as suggested by Theorem 3) to classify nodes as homophilic vs. heterophilic via a simple hard threshold. Using the graph homophily ratio as the threshold, the lowest 81% of nodes by 1-hop feature distance on Cora and 25% on Chameleon are labeled as homophilic. This simple classifier achieves 69.53% accuracy on Cora and 73.25% on Chameleon, substantially higher than random and strongly indicative that feature–neighbor distance encodes meaningful structure in practice.
>
> Node-MoE further strengthens this signal by incorporating 0-hop, 1-hop, and 2-hop feature deviations as inputs to the gating model. The empirical ablation below shows that richer local-structure signals consistently improve performance:
> | | Cora | Chameleon |
> |-|-|-|
> |$[X]$ |89.04 ± 1.38|73.08 ± 2.11|
> |$[AX-X]$|89.10 ± 1.14|73.40 ± 2.07|
> |$[X, AX-X]$| 89.19 ± 1.27 |73.26 ± 1.76|
> |$[X, AX-X, A^2X-X]$|89.38 ± 1.26|73.64 ± 1.80|

---

> ### Author Response · Authors · 2025-11-22
> **Response to Reviewer Jq48 - Part 2**
>
> > W3 & Q3: Building on the point around heuristics, the other core justification the authors make for their method is moving beyond post-fusion/fixed-filters used by previous methods. Specifically, the authors claim that the advantage of their method comes from their node-wise filtering. However, I feel like, again, this is driven by heuristics, rather than a fundamental difference to previous methods. Within Node-MoE, the authors still use soft-routing which leverages all filters, but later apply a top-K gating to decide which to keep. There is no clear justification for why this would work, and feels like a heuristic-based choice that is not sufficiently justified over previous methods.
>
> **A3**: We thank the reviewer for the question. We clarify an important misunderstanding: **Node-MoE does not apply all filters before routing**, and therefore does not perform post-fusion like previous methods. he gating model operates before any filtering and relies only on neighbor–feature distance (0-hop, 1-hop, 2-hop) as input based on the Theorem 3. No spectral filters or expert outputs are computed during gating. Once the gating model determines the routing, only the selected expert (Top-1 gating) is applied, which makes Node-MoE fundamentally different from post-fusion methods, where every node must compute all filters and only mix them afterward.
>
> Moreover, because only one expert is executed per node, Node-MoE avoids redundant computation. As shown below, the runtime (second) remains close to that of a single ChebNetII model:
>
> | Dataset | ChebNetII | Node-MoE (3 experts) | Node-MoE (5 experts) |
> |---|---|---|---|
> | ogbn-arxiv | 1.57 | 1.77 | 1.93 |
> | pokec | 15.58 | 16.6 | 16.79 |
>
> >  W4 & Q4: First, in figure 5, the authors show that the average weights change across homophily level. Yet, expert 1 (the high-pass filter) receives the highest scores for all bins. If this is the case, what is the point of the other expert? I also feel the authors did not provide enough justification for figures 4 and 5. For instance, on line 391 the authors say "This pattern confirms our design that nodes with varying structural patterns require different filters, demonstrating the effectiveness of the proposed gating model", yet this is the premise -- we already know nodes require different filters. I would expect an actual takeaway to show that Node-MoE effectively exploits this information and can adequately route nodes to different models that are aligned with their needs. If we were to take my point above about the weights and the dominance of expert 1, it makes it seem like routing is not actually happening.
>
> **A4**: We thank the reviewer for the helpful comments. We clarify the interpretation of Figures 4 and 5 and provide additional evidence that Node-MoE performs meaningful, homophily-aligned routing.
>
> (1) The learned filters are not strictly "low-pass" and "high-pass".
> Although filter 0 and filter 1 are initialized as low-pass and high-pass, respectively, the learned filters become data-adaptive. As shown in Figure 4, the filter labeled “high-pass’’ still exhibits a noticeable low-frequency response after training. Hence, it is reasonable for expert 1 to contribute non-negligibly even in high-homophily regions.
>
> Importantly, the average soft weight of the "high-pass" expert (Expert 1) decreases as node homophily increases, which implies that the model increasingly relies on the "low-pass" expert (Expert 0) for highly homophilic nodes. This trend is consistent with the intuition that more homophilic regions benefit from smoother (low-pass) filtering.
>
> (2) Top-1 gating reveals clear routing differences across homophily bins.
> Figure 5 visualizes the soft weighted sum of all experts under soft-gating. We further analyze the expert selection with Top-1 gating, where each node only select 1 expert.
>
> We calculate the Top-1 expert selection ratios across different homophily bins on the Chameleon dataset as follows:
>
> | Homophily Range | Expert 0 | Expert 1 |
> | --------------- | -------- | -------- |
> | [0.00, 0.20]    | 0.4310   | 0.5690   |
> | [0.20, 0.40]    | 0.2100   | 0.7900   |
> | [0.40, 0.60]    | 0.3066   | 0.6934   |
> | [0.60, 0.80]    | 0.1982   | 0.8018   |
> | [0.80, 1.00]    | 0.0602   | 0.9398   |
>
> From these results, we observe clear routing differences across homophily ranges: Expert 0 is selected predominantly in low-homophily regions, and its selection probability decreases to nearly zero as the node homophily ratio increases. Conversely, Expert 1 becomes increasingly dominant in high-homophily regions, indicating that the gating network assigns nodes to different experts in a manner that reflects their structural patterns.

---

> ### Author Response · Authors · 2025-11-22
> **Response to Reviewer Jq48 - Part 3**
>
> > W5 & Q5: For figure 6, what is the performance of the more difficult regions, like the heterophilous nodes of Cora and homophilous nodes of Chameleon, when using an MLP? One concern I have is that the GNN models are known to under-perform a basic MLP in these regions, and it possible that simply using an MLP in these regions can outperform Node-MoE. Moreover, I would be interested to see Mowst for Figure 6 given it routes between GNN and MLP. The more fundamental question I am asking is whether Node-MoE actually is leveraging the graph structure in these other areas, or simply choosing to rely less on it.
>
> **A5:** Thank you for the great suggestion. We report the test accuracy across homophily bins for MLP, Mowst, and Node-MoE on both Cora and Chameleon.
>
> | Cora | 0.00-0.20 | 0.20-0.40 | 0.40-0.60 | 0.60-0.80 | 0.80-1.00 |
> |--------|-----------|-----------|-----------|-----------|-----------|
> | # Nodes | 32 | 31 | 43 | 75 | 394 |
> | MLP    | 0.3750 | 0.3226 | 0.5581 | 0.6667 | 0.8579 |
> | Mowst  | 0.1875 | 0.3871 | 0.5814 | 0.9333 | 0.9746 |
> | Node-MoE    | 0.3438 | 0.4194 | 0.7209 | 0.9333 | 0.9898 |
>
> | Chameleon | 0.00-0.20 | 0.20-0.40 | 0.40-0.60 | 0.60-0.80 | 0.80-1.00 |
> |--------|-----------|-----------|-----------|-----------|-----------|
> | # Nodes | 216 | 179 | 53 | 22 | 22 |
> | MLP    | 0.4306 | 0.5307 | 0.3962 | 0.6818 | 0.5909 |
> | Mowst  | 0.5648 | 0.6760 | 0.6604 | 0.7727 | 0.8636 |
> | Node-MoE | 0.7500 | 0.7207 | 0.7358 | 0.8182 | 0.7727 |
>
> From the result, we can find:
>
> * Node-MoE consistently outperforms both MLP and Mowst in most homophily ranges on both datasets, demonstrating that it effectively adapts to different structural patterns.
>
> * On Cora (0.00–0.20), MLP slightly outperforms Node-MoE. However, Node-MoE still outperforms Mowst. This indicates that Mowst fails to route nodes to MLP in this difficult low-homophily region and instead relies too heavily on GCN, while Node-MoE is able to recover reasonable performance.
>
> * On Chameleon (0.80–1.00), Mowst outperforms both Node-MoE and MLP. This again shows that Mowst mainly relies on GCN, which performs well for this small set of highly homophilic nodes. In contrast, Node-MoE learns its own adaptive structural patterns rather than mimicking a pure MLP or pure GCN, resulting in strong performance across all bins.
>
> Overall, these results demonstrate that Node-MoE does not simply avoid using graph structure nor does it collapse to MLP behavior. Instead, it leverages graph structure and outperforms both feature-only (MLP) and model-switching (Mowst) baselines across the majority of structural regimes.
>
> We hope that we have addressed the concerns in your comments, and please kindly let us know if there is any further concern, and we are happy to clarify.

---

### Official Review · Reviewer_vf9y · 2025-10-31

**Soundness:** 3
**Presentation:** 3
**Contribution:** 2
**Rating:** 2
**Confidence:** 4

**Summary:**

This paper studies the node classification problem. It proposes to address the problem where different part of the graph has different homophily. Its core idea is to apply the mixture-of-experts (MOE) framework on it so that every node would be encoded by different filters.

**Strengths:**

S1. The whole framework is intuitive and easy to understand.

S2. The motivation of the proposed method makes sense.

S3. The experimental part compares various datasets and methods.

**Weaknesses:**

W1. The novelty of the proposed method is low. The idea of "applying different filters to different nodes" has been used widely years ago, for example in the model FAGCN [1] , ACM-GCN [2], ALT [3] where every node's filter is tailored. E.g., Figure 3 from this paper and Figure 2 from ALT [3] are quite similar.

W2. I think this paper does not report the best performance of baseline methods. E.g., for ACM-GCN, please check its table 2 from (https://arxiv.org/pdf/2210.07606), which achieves 89.75% on Cora. Also, other competitors from (https://arxiv.org/pdf/2210.07606) seems very capable, even better than the proposed method, in this paper.

[1] Bo, Deyu, et al. "Beyond low-frequency information in graph convolutional networks." Proceedings of the AAAI conference on artificial intelligence. Vol. 35. No. 5. 2021.

[2] Luan, Sitao, et al. "Revisiting heterophily for graph neural networks." Advances in neural information processing systems 35 (2022): 1362-1375.

[3] Xu, Zhe, et al. "Node classification beyond homophily: Towards a general solution." Proceedings of the 29th ACM SIGKDD Conference on Knowledge Discovery and Data Mining. 2023.

**Questions:**

Please check the weaknesses I mentioned.

---

> ### Author Response · Authors · 2025-11-22
> **Resposne to Reviewer vf9y - Part 1**
>
> Dear Reviewer vf9y,
>
> We appreciate your constructive feedback. We are pleased to provide detailed responses to address your concerns.
>
> > W1: The novelty of the proposed method is low. The idea of "applying different filters to different nodes" has been used widely years ago, for example in the model FAGCN [1] , ACM-GCN [2], ALT [3] where every node's filter is tailored. E.g., Figure 3 from this paper and Figure 2 from ALT [3] are quite similar.
>
>
> **A1:** Thank you for raising this concern. While FAGCN, ACM-GCN, and ALT also employ multiple filters, their core mechanisms are fundamentally different from Node-MoE. For clarity, we first summarize their ideas and how they fuse filters:
>
> * FAGCN computes low- and high-frequency components for every node and then fuses them using a learned mixing weight derived from the filtered node representations.
> * ACM-GCN computes multiple predefined channels (identity, low-pass, high-pass) for all nodes, and then applies node-wise attention coefficients to fuse the channel outputs.
> * ALT decomposes the graph into multiple structural views and applies the same GNN to each view. The branch outputs are then fused by a learned fusion network, again based on filtered representations.
>
> Across these models, the pattern is the same: **Every node passes through all filters/channels, and the model fuses these outputs post-hoc based on the filtered representations.**
>
> This post-fusion paradigm introduces several issues:
>
> (1) Post-fusion cannot reliably choose the correct filter. As illustrated in our toy example (Figure 1), applying global filters to all nodes first causes the filtered representations of nodes with different structural patterns to collapse. Once features become nearly identical, it is extremely difficult for any mixing module to recover the correct filter assignment.
>
> (2) All nodes must be computed by all filters.
> Since all filters are always applied to all nodes, the nodes need to calculate the representations of each filter. This prevents experts from specializing and increases computation linearly with the number of filters.
>
> (3) Lack of theoretical justification. Existing methods do not explain when and why multiple distinct filters are required, nor do they provide a principled rule for filter selection.
>
>
> However, Node-MoE is not a post-fusion model. It introduces an expert-routing architecture with several key differences:
>
> **1. Theory-driven, pre-filter gating.**
>
> Routing is performed before applying filters, avoiding the feature-collapse problem and enabling experts to specialize. Each node activates only one expert (Top-1), leading to efficiency comparable to a single-expert model:
>
>  For convenience, we summarize the training time below (in seconds per epoch):
>
> | Dataset | ChebNetII | Node-MoE (3 experts) | Node-MoE (5 experts) |
> |---|---|---|---|
> | ogbn-arxiv | 1.57 | 1.77 | 1.93 |
> | pokec | 15.58 | 16.6 | 16.79 |
>
>
>
> **2. Community-aware routing.**
>
> As shown in our preliminary analysis (Figure 2), nodes in the same community tend to share similar structural patterns. Our structural-aware gating leverages this observation, neighboring nodes are encouraged to route to the same expert, which is validated empirically in Figure 7 and Table 6.
>
>
>
> **3. Theoretical justification for multiple distinct filters. **
>
> We provide formal analysis under the mixed-CSBM model explaining why mixed homophily/heterophily graphs require multiple distinct filters and why node-wise routing based on neighborhood deviation is appropriate. Prior post-fusion methods do not provide such theoretical grounding.
>
> **4. Interpretability through simple filters.**
>
> Node-MoE uses a filter smoothing loss to encourage each expert filter to remain simple and smooth, which helps reduce overfitting and enhances interpretability. As shown in Figures 8, 9, and 10, highly complex filters tend to degrade performance, demonstrating the necessity of controlling filter smoothness. Meanwhile, Figures 11–14 show that the learned filters naturally specialize into meaningful shapes (e.g., low-pass vs. high-pass), and the gating network assigns nodes to these filters in a structurally coherent manner, reflecting the interpretablity of the proposed Node-MoE.
>
> In summary, although prior models combine multiple filters, they all rely on post-fusion mixing, whereas Node-MoE introduces a theoretically motivated expert-routing architecture with distinct specialization, stronger interpretability, and no need to compute all filters for all nodes.

---

> ### Author Response · Authors · 2025-11-22
> **Resposne to Reviewer vf9y - Part 2**
>
> > W2: I think this paper does not report the best performance of baseline methods. E.g., for ACM-GCN, please check its table 2, which achieves 89.75% on Cora. Also, other competitors seems very capable, even better than the proposed method, in this paper.
>
> **A2**: Thank you for pointing out the performance gaps. To ensure a fair comparison across all baselines, we follow the same 10 random splits as in [1, 2] and perform hyperparameter search for every model strictly following their recommended settings. We note that the choice of random split can have a substantial influence on final accuracy, as demonstrated in active learning studies such as [3, 4].
>
> As you suggested, The ACM-GCN in table 2 of ACM-GCN paper achieves 89.75% accuracy on Cora. This result corresponds to the ACM-GCN+ model, which incorporates MLP(A) as additional features. This enhancement is not used by other baselines. For fairness, we therefore initially compared against the standard ACM-GCN without additional features.
>
> To ensure a fair comparison, we extended our approach by incorporating MLP(A) into the experts of Node-MoE, resulting in a variant denoted as Node-MoE+. Using the same experimental setup and hyperparameter tuning (based on the official code of ACM-GCN paper), we compared Node-MoE+ against ACM-GCN+ and ACM-GCN++. The results show that Node-MoE+ also benefits from the inclusion of MLP(A) on heterophilic datasets, achieving competitive performance improvements. However, on homophilic datasets, both ACM-GCN+ and Node-MoE+ exhibit minimal performance gains.
>
> |  | Cora | CiteSeer | Chameleon | Squirrel |
> |:---:|:---:|:---:|:---:|:---:|
> | ACM-GCN | 88.01 ± 1.26 | 76.52 ± 1.72 | 69.62 ± 1.22 | 57.02 ± 0.79 |
> | ACM-GCN+ | 88.16 ± 1.36 | 77.12 ± 1.69 | 74.12 ± 1.78 | 65.65 ± 1.82 |
> | ACM-GCN++ | 88.85 ± 1.05 | 77.17 ± 1.86 | 73.98 ± 1.91 | 64.74 ± 1.96 |
> | Node-MoE | 89.38 ± 1.26 | 77.78 ± 1.36 | 73.64 ± 1.80 | 62.31 ± 1.98 |
> | Node-MoE+ | 89.41 ± 1.28 | 77.90 ± 1.48 | 75.39 ± 1.34 | 66.30 ± 1.99 |
>
> We hope that we have addressed the concerns in your comments, and please kindly let us know if there is any further concern, and we are happy to clarify.
>
> [1] Pei, Hongbin, et al. "Geom-gcn: Geometric graph convolutional networks." ICLR'20.
>
> [2] Lim, Derek, et al. "Large scale learning on non-homophilous graphs: New benchmarks and strong simple methods." NeurIPS'21
>
> [3] Wu, Yuexin, et al. "Active learning for graph neural networks via node feature propagation.", NeurIPS' 19
>
> [4] Ma, Jiaqi, et al. "Partition-based active learning for graph neural networks.", TMLR' 23

---

### Official Review · Reviewer_vdsZ · 2025-11-03

**Soundness:** 3
**Presentation:** 3
**Contribution:** 2
**Rating:** 4
**Confidence:** 4

**Summary:**

The paper addresses a limitation in Graph Neural Networks (GNNs) by proposing a multi-filter approach. It argues that real-world graphs contain a heterogeneous mix of homophilic and heterophilic patterns, so a single filter is suboptimal. The authors provide theory showing that globally optimized filters can harm performance on mismatched substructures, while node-wise filtering can achieve linear separability under mild conditions.

The main technical contributions are:

- A theoretical analysis under a mixed Contextual Stochastic Block Model (CSBM) demonstrating:

  - Global low-pass filtering yields near-optimal separability on homophilic nodes but induces a lower bound on loss for heterophilic nodes.

  - Node-wise application of different filters achieves linear separability across nodes with high probability.

  - A simple, local structure-based criterion can separate homophilic vs. heterophilic nodes with high probability.

- NODE-MOE, a Mixture-of-Experts framework that assigns filters on a per-node basis via a gating network, incorporating local structural cues for structure-aware expert selection.

- Expert models implemented with learnable spectral filters (e.g., ChebNetII) diversified via distinct filter initializations to encourage specialization.

- A filter smoothing loss that reduces spectral oscillations of learned filters to improve trainability and interpretability.

- A Top-K gating variant that activates only the most relevant experts per node, improving efficiency while preserving accuracy.

- Extensive experiments across various datasets show consistent improvements over baselines and MoE alternatives.

**Strengths:**

- Originality: The paper reframes multi-filter GNNs as a node-wise filtering problem, grounding the gating in local structural signals derived from theory. This surpasses post-fusion methods that rely solely on node embeddings.

- Quality: Theoretical results (Theorems 1–3) are coherent, well-connected to the design, and address why global filters fail in mixed regimes. The broad empirical study covers diverse datasets, multiple baselines, and MoE variants. Ablations on gating backbones, Top-K gating, experts, smoothing loss, and noise add credibility.

- Clarity: The problem setting and motivation are well-laid out, with a toy example contrasting global vs. node-wise filtering. The method is explained in modular components with intuitive rationale. Visualizations of learned filters and gating weights aligned to homophily are helpful.

**Weaknesses:**

- Theoretical scope and assumptions:

  - The analysis relies on mixed CSBM and linear classifiers, these restrict direct applicability to modern deep GNNs with nonlinearities.

  - The transition from Theorem 3’s distance criterion to the specific gating input ‎`[X, |AX − X|, |A^2X − X|]` is plausible but not formally tied.

- Methodological clarity:

  - Inconsistency on the expert backbone: Section 4 positions ChebNetII as the expert; Appendix C.4 says “we adopt GCNII as the experts.”

  - Load balancing and Top-K gating details: The paper notes use of a Shazeer-style load-balancing loss but the exact form and its influence are not deeply analyzed.

  - The explanation of the use of GIN as gating model “neighboring nodes are likely to receive similar expert selections” is not well-grounded. Also the details of the GIN model is not given.

  - The ability of ChebNetII to learn diverse filter pattern largely depends on the chosen polynomial order, also Chebyshev polynomial has limitations in approximating nonsmooth functions, e.g. band-pass filters. This limitation is not discussed. Ideally it should be compared with other learnable spectral filters such as ARMA.

- Complexity
  - The filter smoothing loss can be expensive to compute given it requires eigenvalues.

- Presentation
  - The text on figures are too small and hard to read

**Questions:**

- Theorem 3 motivates a local feature-neighborhood distance. Could you formalize why two-hop differences $|A^2X-X|$ add complementary signal beyond one-hop in the gating? Any empirical evidence that 3+ hops saturate or hurt?

- What is the precise form of your load-balancing loss in Top-K gating, and how sensitive are routing distributions to its weight? Please include routing statistics (utilization per expert, imbalance ratio) and its relation to accuracy.

- The smoothing loss curbs spectral oscillations. Why is filter smoothness favored? And does the loss requires explicitly computed eigenvalues?

- In mixed-community graphs, does the gating align with detected communities? If you run community detection and aggregate gating decisions per community, do you observe coherent expert specialization? This could strengthen the interpretability story.

- How does NODE-MOE compare to a “hard assignment” baseline where nodes are first classified as homophilic/heterophilic by a heuristic (e.g., $||X_i - \frac{1}{D_{ii}}\sum_{j\in N(i)}X_j||$ threshold) and then routed to fixed filters? This would test the value of end-to-end learned gating and learnable filters.

- Why and how do you use GIN as gating model? How much layers of GIN you used? If you used layers > 1, theoretically the GIN model can also learn the 1-hop and 2-hop differences in the input, so are the 1-hop and 2-hop differences still needed in the input?

---

> ### Author Response · Authors · 2025-11-22
> **Response to Reviewer vdsZ - Part 1**
>
> Dear Reviewer vdsZ,
>
> We appreciate your constructive feedback. We are pleased to provide detailed responses to address your concerns.
>
> > W1: The analysis relies on mixed CSBM and linear classifiers, these restrict direct applicability to modern deep GNNs with nonlinearities.
>
> **A1:** (Mixed-)CSBM has been widely used in theoretical studies of graph filtering behavior in GNNs [1, 2, 3]. Under this assumption, Theorem 1 shows that applying a single global filter causes the filtered node features of different classes to collapse, making the class-conditional distributions indistinguishable. Once this collapse occurs, no classifier, including nonlinear and multi-layer models, can recover separability because the two distributions are identical after filtering.
>
> In contrast, when we apply different filters to nodes exhibiting different structural patterns, the resulting filtered features remain distinguishable under the mixed-CSBM model. In this case, the two class distributions become separable, and even a linear classifier is sufficient to recover them. This theoretical insight highlights the necessity of node-wise filtering and directly motivates our NODE-MOE framework.
>
> [1] "Graph convolution for semi-supervised classification: Improved linear separability and out-of-distribution generalization", ICML'21
>
> [2] "Is homophily a necessity for graph neural networks?", ICLR'22.
>
> [3] "Demystifying structural disparity in graph neural networks: Can one size fit all?", NeurIPS'23
>
>
> > W2 & Q1 & Q6: The transition from Theorem 3's distance criterion to the specific gating input $[X, |AX - X|, |A^2X - X|]$ is plausible but not formally tied. Any empirical evidence that 3+ hops saturate or hurt? How much layers of GIN you used? If you used layers > 1, theoretically the GIN model can also learn the 1-hop and 2-hop differences in the input, so are the 1-hop and 2-hop differences still needed in the input?
>
> **A2:** We thank the reviewer for the helpful question. Theorem provides a theoretically grounded criterion under the mixed-CSBM assumption, showing that the deviation between a node’s feature and the aggregated features of its neighbors is informative for distinguishing structural patterns. Real-world graphs, however, do not strictly follow the mixed-CSBM generative process, and relying solely on the exact theoretical distance may overlook additional local information present in practical datasets. In most real-world graphs, one-hop and two-hop neighborhoods carry the majority of structurally relevant signals for classification. For this reason, we extend the criterion in Theorem 3 to second-order neighborhoods and use the input $[X,\ |AX - X|,\ |A^{2}X - X|]$ to capture these richer local variations.
>
> In our experiments, we use a 2 layer GIN model as gating model. In the following, we provide empirical comparisons of different gating inputs on the Cora and Chameleon datasets.
>
> | | Cora | Chameleon |
> |---|---|---|
> | $[X]$ | 89.04 ± 1.38 | 73.08 ± 2.11 |
> | $[AX-X]$ | 89.10 ± 1.14 | 73.40 ± 2.07 |
> | $[X, AX-X]$ | 89.19 ± 1.27 | 73.26 ± 1.76 |
> | $[X, AX-X, A^2X-X]$ | 89.38 ± 1.26 | 73.64 ± 1.80 |
> | $[X, AX-X, A^2X-X, A^3X-X]$ | 89.26 ± 1.08 | 73.66 ± 2.14 |
>
>
> From these results, we observe two key patterns: (1) including the 1-hop and 2-hop differences clearly improves performance compared to using $X$ alone, even though a multi-layer GIN could theoretically learn similar quantities; and (2) incorporating 3-hop distances yields no further significant improvment, leading to saturation.
>
>
> > W3 & W8: Inconsistency on the expert backbone: Section 4 positions ChebNetII as the expert; Appendix C.4 says “we adopt GCNII as the experts.” The text on figures are too small and hard to read
>
> **A3**: We thank the reviewer for pointing out this inconsistency. In the main results, we use ChebNetII as the expert model. In Appendix~D.1, we additionally experiment with different GNN architectures as experts to demonstrate the flexibility of NODE-MOE. We will update the appendix C.4 and enlarge the text size in the figures in our revision.

---

> ### Author Response · Authors · 2025-11-22
> **Response to Reviewer vdsZ - Part 2**
>
> > W4 & Q2: Missing Load balancing and Top-K gating details. What is the precise form of your load-balancing loss in Top-K gating, and how sensitive are routing distributions to its weight? Please include routing statistics (utilization per expert, imbalance ratio) and its relation to accuracy.
>
> **A4**: We apply the load-balancing loss in the following form:
> $$\begin{gathered}\text { Importance }(X)=\sum_{x \in X} G(x) \\ L_{\text {balance }}(X)= w*C V(\text { Importance }(X))^2\end{gathered},$$
> where G(x) is the gating weight of sample x, $w$ is the weight of loss, and the loss penalizes the squared coefficient of variation to encourage more even expert utilization.
>
> We conduct experiments on Cora and Chameleon using Top-1 gating with different $w$. Routing distributions and accuracy are shown below.
>
> Cora
> | weight | Expert 0 | Expert 1 | Expert 2 | Accuracy |
> |:---:|:---:|:---:|:---:|:---:|
> | 0 | 1 | 0 | 0 | 88.60 ± 0.55 |
> | 0.001 | 1 | 0 | 0 | 88.91 ± 0.28 |
> | 0.01 | 99.08 | 0.026 | 0.066 | 88.97 ± 0.49 |
> | 0.1 | 78.55 | 21.34 | 0.11 | 89.52 ± 0.18 |
> | 1 | 44.39 | 42.94 | 11.67 | 88.54 ± 0.21 |
>
> Chameleon
>
> | weight | Expert 0 | Expert 1 | Expert 2 | Accuracy |
> |:---:|:---:|:---:|:---:|:---:|
> | 0 | 99.96 | 0.04 | 0 | 71.27 ± 0.76 |
> | 0.001 | 99.96 | 0.04 | 0 | 72.30 ± 0.67 |
> | 0.01 | 51.82 | 40.97 | 7.2 | 73.10 ± 1.28 |
> | 0.1 | 44.44 | 38.21 | 17.35 | 72.51 ± 1.10 |
> | 1 | 39.26 | 32.32 | 28.41 | 70.98 ± 0.77 |
>
>
> These results show that very small weight $w$, such as 0 and 0.001, leads to a collapse into a single expert, causing suboptimal performance. Conversely, very large $w$, such as 1, forces overly uniform routing, which also hurts performance because most nodes belong to the same structural pattern. Moderate balancing (e.g., w=0.01, and w=0.1) yields the best trade-off, preventing collapse while preserving meaningful specialization.
>
> > W5: The explanation of the use of GIN as gating model “neighboring nodes are likely to receive similar expert selections” is not well-grounded. Also the details of the GIN model is not given.
>
>
> **A5**: The aggregation of GIN is shown as follows:
>
> $h_v^{(k)}=\operatorname{MLP}^{(k)}\left(\left(1+\epsilon^{(k)}\right) \cdot h_v^{(k-1)}+\sum_{u \in \mathcal{N}(v)} h_u^{(k-1)}\right)$,
> where $\epsilon^{(k)}$ is a learnable parameter. In our experiments, we use a 2-layer GIN model as the gating model.
>
> As shown in Figure 2 of our paper, real graphs contain local structural regions, where nodes exhibit similar homophily/heterophily patterns. Although neighboring nodes may have different labels or features, they often share similar structural behavior, such as consistently connecting to dissimilar nodes. In these cases, assigning similar experts to neighbors is desirable.
>
> GIN is well-suited for this because its injective aggregation captures local structural coherence, allowing the gating network to learn region-level patterns rather than relying solely on node features. We also evaluated alternative gating architectures, and as shown in Table 6, GNN-based gating, such as GIN, consistently outperforms MLP gating, supporting this design choice.
>
> > W6: The ability of ChebNetII to learn diverse filter pattern largely depends on the chosen polynomial order, also Chebyshev polynomial has limitations in approximating nonsmooth functions, e.g. band-pass filters. This limitation is not discussed. Ideally it should be compared with other learnable spectral filters such as ARMA.
>
> **A6**: Our goal in NODE-MOE is not to rely on a single expert to
> learn arbitrarily complex or nonsmooth filters; instead, we intentionally expect each ChebNetII expert to learn a **simple and smooth** filter. The diversity of filtering behavior arises from the mixture of multiple experts, rather than from a high-order polynomial within a single  expert. This design choice aligns with our theoretical insight that different structural patterns (homophilic vs.\ heterophilic) require distinct but relatively simple filters, rather than complex band-pass shapes.
>
> Regarding the expressiveness of Chebyshev polynomials, we agree that a single Chebyshev polynomial may have limitations in approximating nonsmooth filters. However, NODE-MOE mitigates this by combining multiple experts, initialized with different filter types, which together can represent more diverse spectral behaviors. This is precisely the motivation behind using a mixture-of-experts architecture.
>
> Finally, we also evaluate learnable polynomial filters. As shown in Table 5 of our paper, NODE-MOE still outperforms single complex filters, indicating that the MoE design, rather than the specific filter basis, is the key contributor to performance.

---

> ### Author Response · Authors · 2025-11-22
> **Response to Reviewer vdsZ - Part 3**
>
> > W7 & Q3: The filter smoothing loss can be expensive to compute given it requires eigenvalues. The smoothing loss curbs spectral oscillations. Why is filter smoothness favored? And does the loss requires explicitly computed eigenvalues?
>
> **A7:** We thank the reviewer for raising this concern. In our implementation, the filter smoothing loss does **not** require computing any eigenvalues of the graph Laplacian. ChebNetII parameterizes the spectral filter using Chebyshev polynomial bases $T_k(\cdot)$ evaluated on a fixed set of sampled points in the interval \([0,2]\). These sampling points are predefined and $independent of the graph$, so the smoothing loss is computed entirely in this parameterized spectral domain without performing any eigendecomposition. As a result, the smoothing loss adds negligible overhead to training.
>
> Filter smoothness is favored for two reasons. (1) **Preventing overfitting**: highly oscillatory filters tend to fit noise and lead to overfitting, as shown in Figure 8 of our paper. Encouraging smoothness keeps the learned filters closer to the simple spectral shapes that generalize well. (2) **Improving interpretability**: smoother filters make it easier to interpret the behavior of each expert, enabling clearer understanding of how NODE-MOE adapts to different structural patterns.
>
> > Q4: In mixed-community graphs, does the gating align with detected communities? If you run community detection and aggregate gating decisions per community, do you observe coherent expert specialization? This could strengthen the interpretability story.
>
> **A8**: Thanks for the insightful suggestion. We follow the reviewer’s idea and analyze whether the learned gating aligns with graph communities. We run community detection and compute, for each community, the consistency, defined as the fraction of nodes assigned to the community’s most common expert. Experiments are conducted on Cora and Chameleon using Top-1 gating with 3 experts.
>
> Cora
>
> | Community ID | 0 | 1 | 2 | 3 | 4 | 5 | 6 | 7 | 8 | 9 |
> |--------------|---|---|---|---|---|---|---|---|---|---|
> | Size | 404 | 391 | 212 | 199 | 171 | 149 | 115 | 107 | 106 | 101 |
> | Consistency | 0.8457 | 0.8704 | 0.7531 | 0.8827 | 0.9747 | 0.9038 | 0.9710 | 0.8037 | 0.9371 | 0.7888 |
>
> The overall expert distribution is [78.55, 21.34, 0.11], while weighted average consistency is 86.71\%, which is larger than the 78.55\%.
>
> Chameleon
>
> | Community ID | 0 | 1 | 2 | 3 | 4 | 5 | 6 | 7 | 8 | 9 |
> |--------------|---|---|---|---|---|---|---|---|---|---|
> | Size | 404 | 391 | 212 | 199 | 171 | 149 | 115 | 107 | 106 | 101 |
> | Consistency | 0.7616 | 0.7442 | 0.7327 | 0.7420 | 0.8265 | 0.8523 | 0.8812 | 0.6822 | 0.7987 | 0.6040 |
>
> The overall expert distribution is [51.82, 40.97, 7.2], while weighted average consistency is 77.23\%, which is larger than the 51.82\%.
>
> These results show that nodes within the same community tend to be routed to the same expert, and the consistency within communities is significantly higher than the global expert proportions. This provides direct evidence that NODE-MOE captures community-level structural patterns, strengthening the interpretability of the gating mechanism.
>
> > Q5: How does NODE-MOE compare to a “hard assignment” baseline where nodes are first classified as homophilic/heterophilic by a heuristic (e.g., $\left\|X_i-\frac{1}{D_{i i}} \sum_{j \in N(i)} X_j\right\|$ threshold) and then routed to fixed filters? This would test the value of end-to-end learned gating and learnable filters.
>
> **A5:** Thanks for the helpful suggestion. We construct a hard-assignment baseline following the reviewer’s idea. We first compute the node-level homophily score using the 1-hop feature distance and threshold it using each graph’s homophily ratio. This assigns nodes to homophilic or heterophilic groups (e.g., 81% homophilic in Cora, 25% in Chameleon). We evaluate three settings: (1) **low–high**: homophilic nodes → fixed low-pass filter; heterophilic nodes → fixed high-pass filter; (2) **high–low**: inverse of 1, where homophilic nodes → fixed high-pass filter; heterophilic nodes → fixed low-pass filter; (3) **learnable filters**: same routing as 1, but filters are learnable instead of fixed.
>
> | Hard Assignment |  low-high | high-low | learnable filters | Node-MoE |
> |:---:|:---:|:---:|:---:|:---:|
> | Cora | 78.68 ± 0.86 | 56.07 ± 1.73 | 87.90 ± 1.28 | 89.19 ± 1.53 |
> | Chameleon | 55.02 ± 2.25 | 52.41 ± 1.91 | 70.15 ± 1.38 | 73.55 ± 1.74 |
>
> These results show three key findings:
>
> * The 1-hop distance heuristic is meaningful: low–high clearly outperforms high–low, especially on Cora.
> * Hard assignment with fixed filters is suboptimal, while allowing the filters to be learnable improves performance.
> * NODE-MOE further improves over hard assignment, demonstrating the value of end-to-end learnable gating model and the flexibility of learnable experts.
>
> Please kindly let us know if there is any further concern, and we are happy to clarify.

---

### Meta-Review · Area_Chair_YK7g · 2025-12-05

**Summary:**

Reviewers’ concerns focus on methodological clarity,  result analysis,  novelty, performance improvement,  and the role of the theorems. Some are partially answered,  while I think some critical ones are still remarkable.  Although none of the reviewers participated in the discussion, I think they will maintain the score. Thus,  I tend to reject this paper.

**Reviewer Concerns:**

I think that both the methodological clarity (Reviewer vdsZ) and result analysis (Reviewer Jq48) are partially alleviated.  However, some critical concerns are not properly answered.  Firstly, the novelty of the proposed method is limited, as pointed out by Reviewer vf9y with sufficient reference.  Secondly, the performance improvement is marginal. The authors’ responses do NOT include results on large graphs. Thirdly, the direct relationship between theorems and the provided methods is unclear.

**Reviewer Scores:**

Although none of the reviewers participated in the discussion, I think most of them will maintain the score, since some main concerns are not properly answered. Thus, I think the final scores may be as follows.
- Jq48	Rating: 4 / Confidence: 4
- vdsZ	Rating: 4 / Confidence: 4
- vf9y	Rating: 2 / Confidence: 4

---

### Decision · Program_Chairs · 2026-01-26

Reject